# Differential Smoothing Mitigates Sharpening and Improves LLM Reasoning

## Abstract

It is widely recognized that reinforcement learning (RL) fine-tuning of large language models often leads to *diversity collapse*, where outputs lack variety. Specifically, RL tends to amplify existing proficiencies (on tasks it performs well) rather than rectify initial deficiencies (on tasks it struggles with). Prior work has proposed a range of heuristics to counteract this effect, but these methods are ad hoc: they frequently trade off correctness for diversity, their effectiveness varies across tasks, and in some cases they even contradict one another. In this work, we place these observations on a rigorous foundation. We first provide a formal proof of why RL fine-tuning exhibits diversity collapse. Building directly on this analysis, we introduce a principled method—*differential smoothing*—that provably improves both correctness and diversity, outperforming vanilla RL as well as widely used entropy-based heuristics. Our theory precisely characterizes when existing heuristics help and why they fail, while showing that differential smoothing is universally superior. Extensive experiments with models from 1B to 7B parameters, across domains including CountDown and real-world mathematical reasoning, demonstrate consistent gains. Differential smoothing improves both Pass@1 (correctness) and Pass@k (diversity), with up to 6.7% improvements on AIME24 dataset.

## 1 Introduction

Reinforcement learning (RL) has become a powerful technique for fine-tuning Large Language Models (LLMs), enhancing capabilities ranging from complex reasoning (Guo et al., 2025; Yu et al., 2025; Shao et al., 2024) to human preference alignment (Ouyang et al., 2022; Bai et al., 2022a). However, this process is often plagued by a significant side effect: a collapse in generation diversity (Song et al., 2024; Dang et al., 2025b; Yue et al., 2025; Zhao et al., 2025; He et al., 2025a). This degradation is empirically observed in metrics like Pass@$K$; RL-tuned models often show diminishing improvements for larger values of $K$ and can even underperform the original base model (He et al., 2025a; Cobbe et al., 2021; Chow et al., 2024; Chen et al., 2025b). As output diversity is crucial for downstream applications and performance scaling (Wu et al., 2024; Snell et al., 2024), it is imperative to understand and counteract this effect of diversity collapse.

However, mitigating this diversity collapse is non-trivial and presents several challenges. First, there is a persistent trade-off between correctness and diversity. Simple heuristics such as early stopping or high-temperature decoding may boost diversity and achieve higher Pass@K, but they frequently reduce correctness and hurt Pass@1 performance. Second, most existing methods lack robustness across settings. Our experiments confirm that techniques designed to enhance diversity often succeed only on the tasks for which they were originally developed. A striking example is entropy control, where some works recommend maximizing entropy to improve both Pass@1 and Pass@K, while others report that minimizing entropy can yield the same outcome.

Motivated by these limitations, our work has two primary goals: (1) to develop a principled method that robustly improves both correctness (Pass@1) and diversity (Pass@K) across a range of benchmarks, and (2) to provide clarity on the seemingly contradictory effects of previous methods.

We analyze diversity collapse from first principles in a formal setting in ection 3.2. Our analysis shows that RL fine-tuning introduces two compounding biases that cause diversity collapse. Selection bias arises because correct trajectories that have high-probability under the pre-trained model

are more likely to be reinforced (Theorem B.1), and reinforcement bias arises because these same trajectories receive disproportionately larger updates (Theorem B.2).

Leveraging the insights from our theoretical analysis, we propose a *simple but novel twist* to vanilla RL that is designed to simultaneously enhance correctness (Pass@1) and diversity (Pass@K). The core of our method is that we can mitigate the tradeoff between diversity and correctness by using differentiated reward mechanism that applies distinct pressures to correct and incorrect trajectories.

We propose the **differential smoothing** approach. For correct trajectories, our reward mitigates the diversity collapse by subtracting a term proportional to their log-probability. On incorrect trajectories, our reward modification focuses on correctness, by adding the log-probability of the incorrect trace. We present our proposed DS-GRPO algorithm in Section 4.2.

We validate our differential smoothing approach both theoretically and empirically. Our theoretical analysis (Section 6) formally proves that the reward modification for correct trajectories directly optimizes for diversity, while the adjustment for incorrect ones enhances correctness without compromising diversity.

We evaluate DS-GRPO across a range of real-world settings, from simpler tasks such as Countdown to more challenging benchmarks in mathematical reasoning (MATH500 (Hendrycks et al., 2021), OlympiadBench (He et al., 2024), AMC23 (math ai, 2025), AIME24 (H4, 2025) and AIME25 (OpenCompass, 2025)). Our experiments cover multiple base models (Qwen2.5-Math-1.5B (Qwen Team, 2024), Qwen3-1.7B (Qwen Team, 2025) and Ministral-8B-Instruct (Jiang et al., 2024), Qwen2.5-3B-Instruct(Qwen Team, 2024)), and we consistently observe that DS-GRPO improves both Pass@1 and Pass@K relative to the vanilla baseline. This validates our theory that the method enhances diversity without sacrificing correctness, and vice versa. We further compare against prior heuristics, including entropy regularization and recent diversity-prompting techniques (He et al., 2025a; Chen et al., 2025c; Walder & Karkhanis, 2025a) . While these baselines yield improvements only in certain settings, DS-GRPO delivers robust gains across *all* datasets and models tested. Thus, DS-GRPO represents a principled approach that not only improves upon vanilla RL but also provides consistently stronger results than existing heuristics.

Finally, our analysis clarifies the contradictory effects of global entropy regularization in prior work. Increasing entropy across all trajectories improves diversity but reduces correctness, which can help on tasks with many valid solutions. Conversely, decreasing entropy improves correctness at the cost of diversity, which suits tasks with few valid solutions. Our experiments confirm this principle, and we further show that *differential entropy control*, increasing entropy on correct trajectories while decreasing it on incorrect ones—achieves the best of both, paralleling the effect of differential smoothing.

> **Summary of Our Main Contribution**
>
> 1. We analyze diversity collapse from first principles in a formal setting.
> 2. Based on our diagnosis, we propose a novel differential smoothing algorithm that empirically improves both Pass@1 and Pass@K and outperforms previous methods *robustly* in various real-world settings.
> 3. We formally prove that our proposed differential smoothing approach improves diversity and correctness over the vanilla approach and the popular entropy maximization heuristic.
> 4. The analysis in this work also has broader implications of clarifying when and why existing heuristics work and guide the principled modifications of such heuristics.

## 2 RELATED WORK

**Mitigating Diversity Collapse in RL for Reasoning.** Fine-tuning language models with reinforcement learning often causes "diversity collapse", where the policy sharpens around a few solutions (Dang et al., 2025b; Yue et al., 2025). While prior work has empirically documented this effect (Wu et al., 2025), our primary contribution is a formal theoretical framework that rigorously explains its underlying cause. Existing methods to mitigate this issue, such as optimizing for Pass@K (Tang et al., 2025; Walder & Karkhanis, 2025a) or encouraging low-probability solutions (He et al., 2025a; Song et al., 2025), often improve diversity (Pass@K) at the expense of correctness (Pass@1) and

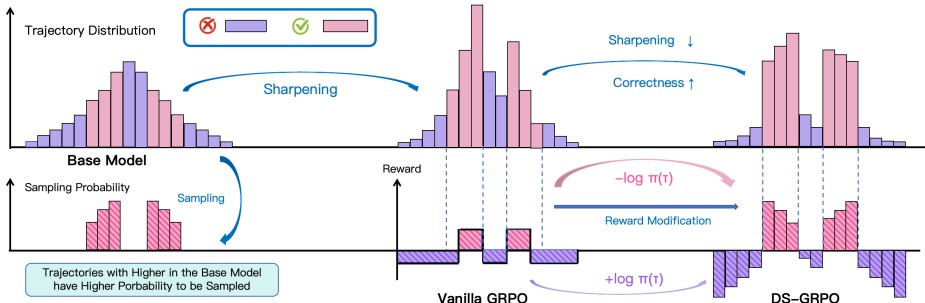

*Figure 1: An illustration of the sharpening effect in vanilla RL and the mitigation mechanism of DS-GRPO*

lack consistent performance across tasks. In contrast, the algorithm derived from our framework is designed to overcome this trade-off. We provide theoretical guarantees and empirical evidence showing that it simultaneously improves both Pass@1 and Pass@K across multiple reasoning benchmarks.

**Controlling Distribution Entropy in RLVR.** Controlling policy entropy is a common technique in RLVR, but its application is debated. Some studies advocate for maximizing entropy to encourage exploration and diversity (Yu et al., 2025; He et al., 2025b; Liu et al., 2025), while others report that minimizing it can improve single-solution accuracy (Agarwal et al., 2025; Gao et al., 2025). This has led to conflicting findings and uncertainty about the optimal strategy. Our work addresses this ambiguity by reframing our method as a novel form of entropy control that outperforms these global strategies. Our analysis clarifies the inconsistent effects of entropy and provides a new principle for its effective regulation.

# 3 THEORETICAL FRAMEWORK FOR THE DIVERSITY COLLAPSE IN RL

In this section, we theoretically analyze the diversity collapse effect that arises during the RL fine-tuning of LLMs. We begin by describing the theoretical abstraction for the RL fine-tuning process (Sec 3.1). We then present two driving factors for diversity collapse (Sec 3.2). Finally, we derive a principled reward function that mitigates the diversity collapse and enhances diversity (Sec 3.3).

## 3.1 SETUP

**Preliminaries.** Our theoretical model for language generation is a token-level Markov Decision Process. The environment is specified by a state space $\mathcal{S}$, a token vocabulary $\mathcal{A}$, a binary reward function $r \in \mathcal{R}$ mapping state-action pairs to $\{0, 1\}$, and a maximum length $H$. Each episode starts with an input problem $\mathbf{x} \in \mathcal{X}$, which defines the initial state $s_1$. The state evolves deterministically based on token selection: at step $h$, the state is $s_h = (\mathbf{x}, a_1, \ldots, a_{h-1})$, and choosing token $a_h$ leads to $s_{h+1} = (s_h, a_h)$. The agent's behavior is described by a policy $\pi$, which provides a distribution over tokens at each state, $\pi_h : \mathcal{S}_h \to \Delta(\mathcal{A})$. We define a trajectory as the full sequence $\tau = (\mathbf{x}, a_1, \ldots, a_H)$, and its cumulative reward is given by $r(\tau)$. We write $\mathbb{D}_{\mathrm{KL}}(\cdot \| \cdot)$ for the KL divergence and $\mathbb{D}_{\chi^2}(\cdot \| \cdot)$ for the $\chi^2$-divergence between two distributions.

**RL fine-tuning over a base policy.** A base model (or a pre-trained model) with corresponding policy $\pi_{\mathrm{base}}$ is fine-tuned to optimize the following objective:

$$\pi_{\mathrm{van}}^*(\tau) = \arg\max_{\pi} \mathbb{E}_{\tau \sim \pi} r(\tau) - \beta \cdot \mathbb{D}_{\mathrm{KL}}(\pi \| \pi_{\mathrm{base}}), \qquad (1)$$

where the KL-divergence term serves as a regularizer that prevents the updated policy from deviating too much from the base policy $\pi_{\mathrm{base}}$ and $\beta$ is a hyperparameter that balances the trade-off between maximizing reward and preserving the knowledge of the base model. We denote this by $\pi_{\mathrm{van}}^*$ to distinguish from proposed improvements in later sections.

In practical applications, an explicit reward function is typically unavailable. Instead, rewards are discovered empirically by sampling potential solutions from a base model and evaluating them with an external verifier. To formalize this process, our theoretical framework defines the reward function through a similar sampling procedure. Initially, all trajectories are assigned a reward of zero. A set of trajectories is then sampled from the base policy, $\pi_{\mathrm{base}}$, and a verifier identifies the successful ones, whose rewards are subsequently set to 1. Consequently, the reward function $r(\tau)$ in Equation 1 is not predetermined but is contingent on the specific set of successful trajectories discovered in the initial sampling phase.

## 3.2 THEORETICAL RESULT ON DIVERSITY COLLAPSE

With the setup in place, we identify two fundamental mechanisms that drive the diversity collapse, captured in the following theorems.

---

**Analysis of Diversity Collapse in RL**

**Proposition 3.1** (Selection Bias)**.** *The probability that a correct trajectory's likelihood increases is monotonically related to its initial probability under the base model. Formally, for any two correct trajectories $\tau_1, \tau_2$ and $\beta > 0$, we have*

$$\pi_{base}(\tau_1) \geq \pi_{base}(\tau_2) \implies \mathbb{P}(\pi_{van}^*(\tau_1) > \pi_{base}(\tau_1)) \geq \mathbb{P}(\pi_{van}^*(\tau_2) > \pi_{base}(\tau_2)).$$

**Proposition 3.2** (Reinforcement bias)**.** *The magnitude of probability gain for a given trajectory is directly proportional to its probability under the base policy. Formally, if the reward update mechanism has access to the complete set of correct trajectories ($r(\tau) = 1$ for all correct trajectories), then for any trajectory $\tau$ and $\beta > 0$, we have*

$$\pi_{van}^*(\tau) - \pi_{base}(\tau) \propto \pi_{base}(\tau).$$

---

Proposition B.1 reveals a *selection bias*: among correct trajectories, those with higher base probabilities are more likely to be reinforced. In addition, Proposition B.2 shows that there is a *reinforcement bias*: these same high-probability trajectories receive disproportionately larger boosts, further amplifying the model's existing preferences and sharpening the distribution. The proofs for Proposition B.2 and Proposition B.1 are provided in Appendix B.1. These results are derived by directly calculating the expression for the fine-tuned policy and analyzing its resulting probability distribution across trajectories.

**Remark 1.** *Proposition B.1 explains the surprising finding of Zhu et al. (2025) that using **only negative samples improves diversity** (Pass@$K$). Using only negative samples mitigates the selection bias, as all positive trajectories implicitly have the same probability of seeing an increase in likelihood over the base policy. This reduces the diversity collapse and improves diversity.*

## 3.3 NEW REWARD FUNCTION TO MITIGATE DIVERSITY COLLAPSE

We have established that vanilla RL induces a **diversity collapse**: high-probability correct responses from the base model are disproportionately reinforced, while low-probability correct responses are neglected. Intuitively, this bias can be countered by reshaping the reward to favor low-probability correct responses.

To do so in a principled manner, we first analytically derive the optimal fine-tuned policy for a given reward function. As shown in Lemma 1, the fine-tuned policy is proportional to the exponentiated reward $\pi^\star(\tau) \propto \exp(r(\tau)/\beta)$.

Guided by this expression, we propose subtracting a term $\gamma_p \cdot \log(\pi_{\text{base}}(\tau))$ from the rewards on correct trajectories, where $\pi_{\text{base}}$ denotes the base policy and $\gamma_p$ is a hyperparameter. We formally prove that this modification mitigates the diversity collapse of vanilla RL in latter sections. In Section 6, we theoretically show that our approach enhances policy diversity. Furthermore, in Appendix E.2, we provide empirical evidence that this diversity improvement is driven by modifying the reward for correct trajectories.

For incorrect responses, however, subtracting such a term is unnecessary since diversity among incorrect outputs is not desirable. On the contrary, *adding* a corresponding term $\gamma_n \cdot \log(\pi_{\text{base}}(\tau))$ can improve correctness, consistent with prevailing intuition that entropy minimization enhances accuracy (Gao et al., 2025; Agarwal et al., 2025). We demonstrate both theoretically in Section 6 and empirically in Appendix E.2 that this modification for incorrect trajectories does not exacerbate the selection or reinforcement biases over correct trajectories.

Putting these together, we propose the **differential smoothing** reward function for a trajectory $\tau$:

$$r_{\text{DS}}(\tau) = \begin{cases} r(\tau) - \gamma_p \cdot \log(\pi_{\text{base}}(\tau)) & \text{if } r(\tau) > 0 \quad \text{(correct trajectories)} \\ r(\tau) + \gamma_n \cdot \log(\pi_{\text{base}}(\tau)) & \text{if } r(\tau) \leq 0 \quad \text{(incorrect trajectories),} \end{cases} \tag{2}$$

where $\gamma_p, \gamma_n \geq 0$ are hyperparameters.

We first investigate the practical benefits of our proposed differential smoothing in Section 4, and theoretically prove the superiority of differential smoothing over vanilla training and existing heuristics in Section 6.

# 4 EXPERIMENTAL ANALYSIS FOR DS-GRPO

In this section, we empirically evaluate the effectiveness of proposed reward modification (Eq.2) in LLM reinforcement finetuning. We show that our method improves both Pass@1 and Pass@K across various tasks and models, outperforming the baseline methods.

## 4.1 PRELIMINARIES: GROUP RELATIVE POLICY OPTIMIZATION

We adopt *Group Relative Policy Optimization* (GRPO) (Shao et al., 2024) as our training backbone. For each input $x$ sampled from the training set, the policy decodes a group of $G$ completions $\{y_i\}_{i=1}^{G} \sim \pi_{\theta_{\text{old}}}(\cdot \mid x)$, where $\pi_{\theta_{\text{old}}}$ denotes the behavior policy used to collect the batch (the policy parameters at the previous update). Let $r_i = r(y_i)$ be the scalar reward of completion $y_i$, and denote by $\mu(\{r_j\}_{j=1}^{G})$ and $\sigma(\{r_j\}_{j=1}^{G})$ the mean and standard deviation of $\{r_j\}_{j=1}^{G}$, respectively. GRPO replaces the learned critic with a *group baseline* and uses the group-standardized advantage

$$A_i = \frac{r_i - \mu(\{r_j\}_{j=1}^{G})}{\sigma(\{r_j\}_{j=1}^{G})}. \tag{3}$$

GRPO adopts a clipped objective with a forward KL regularizer to the fixed *base policy* $\pi^b$:

$$\mathcal{J}_{\text{GRPO}}(\theta) = \mathbb{E}_x \mathbb{E}_{\{y_i\}_{i=1}^{G} \sim \pi_{\theta_{\text{old}}}(\cdot | q)}$$

$$\left[ \frac{1}{G} \sum_{i=1}^{G} \frac{1}{|y_i|} \sum_{t=1}^{|y_i|} \min\left( \rho_{i,t}(\theta) A_i, \text{clip}\left( \rho_{i,t}(\theta), 1-\epsilon, 1+\epsilon \right) A_i \right) - \beta_{\text{KL}} \mathbb{D}_{\text{KL}}\left( \pi_\theta \,\|\, \pi^b \right) \right],$$

where $\rho_{i,t}(\theta) = \frac{\pi_\theta(y_{i,t} | x, y_{i,<t})}{\pi_{\theta_{\text{old}}}(y_{i,t} | x, y_{i,<t})}$ is the importance ratio.

## 4.2 REWARD MODIFICATION FOR MITIGATING DIVERSITY COLLAPSE

To operationalize the theoretical principles from Section 3.3 within the GRPO framework, we propose a novel algorithm, *Differential Smoothing GRPO* (DS-GRPO). Our method works by reshaping the advantage function $A_i$, as specified in Equation equation 4. Specifically, for successful completions (where reward $r_i = 1$), we subtract the term $\gamma_p \log \pi_{\theta_{\text{old}}}(y_i \mid x)$ from the advantage. Conversely, for unsuccessful completions, we add the term $\gamma_n \log \pi_{\theta_{\text{old}}}(y_i \mid x)$.

---

**Differential Smoothing GRPO (DS-GRPO)**

$$A_i^{\text{DS}} = A_i + \begin{cases} -\gamma_p \, \log \pi_{\theta_{\text{old}}}(y_i \mid x), & \text{if } r_i = 1, \\ +\gamma_n \, \log \pi_{\theta_{\text{old}}}(y_i \mid x), & \text{otherwise,} \end{cases} \tag{4}$$

We plug the modified advantages $A_i^{\text{DS}}$ into the GRPO objective:

$$\mathcal{J}_{\text{DS}}(\theta) = \mathbb{E}_x \mathbb{E}_{\{y_i\}_{i=1}^{G} \sim \pi_{\theta_{\text{old}}}(\cdot | q)}$$

$$\left[ \frac{1}{G} \sum_{i=1}^{G} \frac{1}{|y_i|} \sum_{t=1}^{|y_i|} \min\left( \rho_{i,t}(\theta) A_i^{\text{DS}}, \text{clip}\left( \rho_{i,t}(\theta), 1-\epsilon, 1+\epsilon \right) A_i^{\text{DS}} \right) - \beta_{\text{KL}} \mathbb{D}_{\text{KL}}\left( \pi_\theta \,\|\, \pi^b \right) \right].$$

---

We evaluate DS-GRPO on the Countdown and MATH reasoning benchmarks across a range of models. As demonstrated in the subsequent sections, our method consistently improves both correctness (Pass@1) and diversity (Pass@K), outperforming existing diversity-promoting approaches on all evaluation metrics.

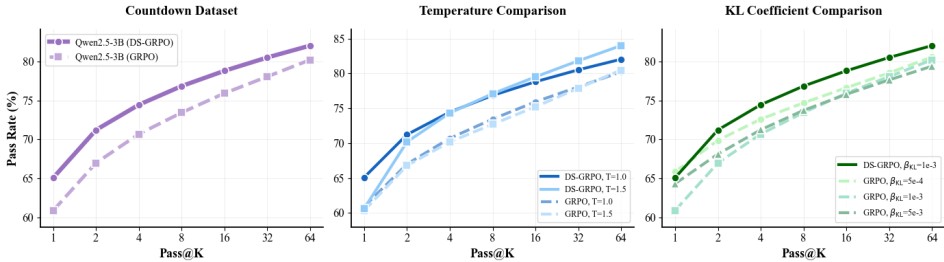

*Figure 2: Pass@K performance of DS-GRPO on the Countdown task, compared with GRPO under varying decoding temperatures and KL coefficients.*

## 4.3 COMPARISON OF DS-GRPO WITH VANILLA GRPO

As shown in Figure 2, DS-GRPO demonstrates remarkable robustness. It consistently enhances Pass@K (for all $K$) by $\approx 4\%$ compared to vanilla GRPO. Crucially, these performance gains are accompanied by a $4\times$ inference speedup. (See Appendix D for additional results).

Figure 3 extends our evaluation to three base models and five mathematical reasoning benchmarks. Our strategy yields substantial improvements, with Pass@1 gains of 0.2%–2.9% and Pass@64 gains of 0.5%–6.7% (detailed in Appendix E.1). This demonstrates our method's ability to improve RL reasoning while mitigating diversity collapse. Furthermore, it delivers significant efficiency gains: DS-GRPO matches the Pass@64 of vanilla GRPO using only $k = 16$ samples—yielding a nearly $4\times$ inference speedup—while simultaneously pushing the maximum achievable Pass@K. This uniform uplift underscores our approach's efficacy in enhancing both exploration and diversity.

## 4.4 ABLATION EXPERIMENTS ON HYPERPARAMETERS

To demonstrate the robustness of our method, we further evaluate performance across different hyperparameters, including sampling temperature, the KL coefficient $\beta_{KL}$, and the reward modification coefficients $\gamma_p$ and $\gamma_n$.

**Temperature and KL Coefficient.** We evaluate the stability of DS-GRPO across varying sampling temperatures and KL coefficients ($\beta_{KL}$). Our results demonstrate consistent improvements over vanilla GRPO: DS-GRPO enhances Pass@K (for all $K$) by $\approx 4\%$ across the temperature range and by $\approx 3.2\%$ across different KL coefficients.

**Reward Modification Coefficient** To isolate the contribution of each component in our reward modification strategy, we conduct an ablation study. We compare the full DS-GRPO algorithm against two specialized variants: *DS-GRPO-Positive*, which only modifies the advantage for correct trajectories, and *DS-GRPO-Negative*, which only modifies the advantage for incorrect trajectories. Their respective advantage modifications are defined as follows:

$$A_i^{\text{DS+}} = A_i - \gamma_p \, \log \pi_{\theta_{\text{old}}}(y_i \mid x), \quad \text{if } r_i = 1, \quad A_i^{\text{DS-}} = A_i + \gamma_n \, \log \pi_{\theta_{\text{old}}}(y_i \mid x), \quad \text{if } r_i \neq 1.$$

The full DS-GRPO algorithm demonstrates superior performance over both of its individual components (DS-GRPO-Positive and DS-GRPO-Negative) for all $K$. Detailed results and discussion are available in Section E.2. sharpening.

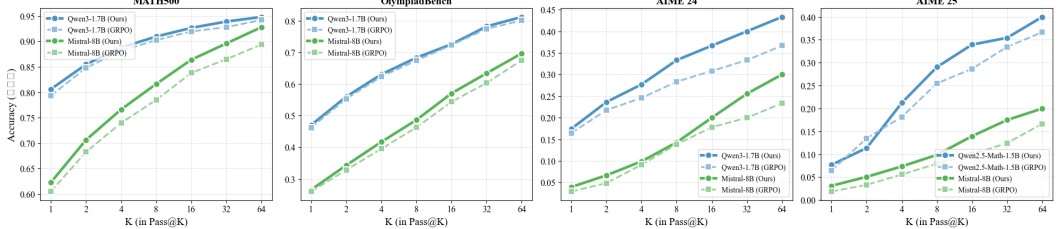

*Figure 3: Pass@K performance after reward modification, compared with vanilla GRPO. X-axis denotes K and y-axis denotes pass rates. Trained on DAPO(Yu et al., 2025) and MATH(Hendrycks et al., 2021) Dataset.*

## 4.5 COMPARISON WITH OTHER METHODS FOR INCREASING DIVERSITY

We compare DS-GRPO with prior approaches that encourage diversity in reasoning through reward or advantage shaping, either by optimizing Pass@$K$ rate directly(Tang et al., 2025; Walder &

Karkhanis, 2025a; Chen et al., 2025c) or by applying rank-based penalties (He et al., 2025a). For mathematical reasoning experiments, we use the Mistral-8B-Instruct(Jiang et al., 2024) as the base model. For PKPO and GR-PKPO, we use $K = 4$ which performs best in previous work (Chen et al., 2025c); for rank-based panelty, we sweep across various configurations. See more result details in Appendix D.

**Pass@K Optimization Methods.** Methods that directly optimize the Pass@$K$ metric use it as a reward signal (Tang et al., 2025; Walder & Karkhanis, 2025a; Chen et al., 2025c). However, this approach can assign zero reward to correct solutions, which increases gradient variance and harms training stability. Experimentally, these methods often trade correctness for diversity; for instance, GR-PKPO slightly improves Pass@64 at the cost of Pass@1 and is unstable on the Countdown task (Figure 4, Left). In contrast, DS-GRPO consistently improves Pass@$K$ across all values of $K$.

**Comparison with Unlikeliness Reward Method.** Our work is conceptually similar to methods that reward unlikely solutions, such as the one proposed by He et al. (2025a). However, our approach has key advantages. DS-GRPO is derived from a theoretical framework that guarantees its optimality. More critically, it employs a differentiated reward strategy: it modifies rewards for correct trajectories to boost diversity, while a complementary modification for incorrect trajectories improves correctness. In contrast, methods like that of He et al. (2025a) focus solely on diversity, which can harm correctness. Our experimental results (Figure 4) validate this, showing DS-GRPO's superior performance across all values of $K$.

**Comparison with Other RL Reasoning Methods.** We further compare our approach with a recently proposed method that focuses on improving RL reasoning: CISPO (Chen et al., 2025a). Empirical results demonstrate that our method consistently outperforms CISPO across all mathematical reasoning datasets. Detailed comparisons are provided in Appendix F.3.

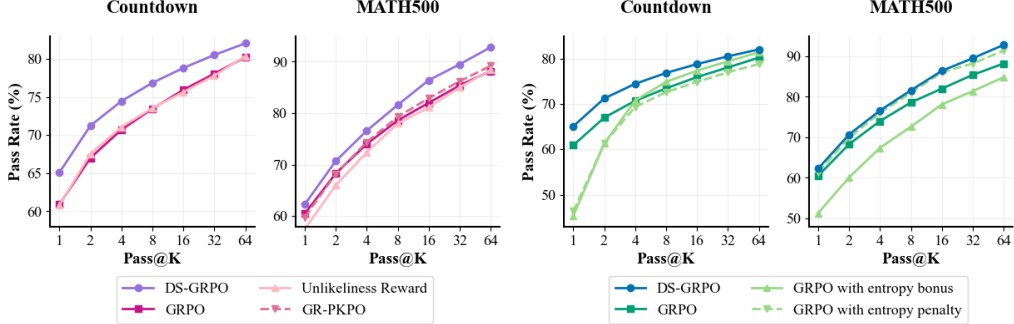

Figure 4: *Performance comparisons on MATH500 and Countdown. Left: Comparison among DS-GRPO, GRPO, GR-PKPO (Chen et al., 2025c), and the Unlikeliness Reward method (He et al., 2025a). Right: Comparison among DS-GRPO, GRPO, Entropy Regularization, and Entropy Minimization. On the Countdown task, training with GR-PKPO collapses, so results are omitted.*

## 5  A DIFFERENTIATED APPROACH TO ENTROPY CONTROL

Doing entropy control is a commonly adapted way of increasing diversity and improving LLM reasoning ability. In this section, we formally compare our method to previous entropy control method from empirical perspective, and in latter section (Section 6) we will theoretically prove that our method outperform vanilla GRPO and GRPO with entropy direct entropy maximization.

### 5.1  DS-GRPO OUTPERFORM ENTROPY BASED METHOD

The role of entropy in RL fine-tuning is complex and subject to ongoing debate. While conventional methods employ entropy regularization to prevent policy collapse (Schulman et al., 2017), recent studies suggest that explicitly *minimizing* entropy can, counter-intuitively, boost performance in certain scenarios (Agarwal et al., 2025; Xingjin Wang, 2025). To position our method within this context, we compare it against two direct entropy control baselines: one that adds an **entropy bonus** to encourage exploration, and one that applies an **entropy penalty** to encourage exploitation. The

respective optimization objectives are:

$$\mathcal{J}_{\text{ent+}} = \mathcal{J}_{\text{GRPO}}(\theta) - \eta_+ \sum_y \pi_\theta(y \mid x) \log(\pi_\theta(y \mid x)), \quad \text{Entropy Bonus}$$

$$\mathcal{J}_{\text{ent-}} = \mathcal{J}_{\text{GRPO}}(\theta) + \eta_- \sum_y \pi_\theta(y \mid x) \log(\pi_\theta(y \mid x)), \quad \text{Entropy Penalty}$$

As illustrated in Fig. 4, our method consistently outperforms both the entropy bonus and penalty approaches, regardless of the direction of the regularization. This suggests that a simple, global adjustment to entropy is less effective than our differentiated reward strategy.

## 5.2 DEEPER DISCUSSION ON ENTROPY CONTROL

Our experiments (Fig. 4) reveal a critical insight: the effectiveness of global entropy control is highly task-dependent. Specifically, an entropy bonus improves performance over vanilla GRPO on the Countdown task but hinders it on math reasoning benchmarks. Conversely, an entropy penalty benefits math reasoning while degrading performance on Countdown. *Can we explain these differing trends?*

**A Principle for Task-Aware Entropy Control.** Global entropy bonus does increase diversity but comes at the cost of correctness. This is part of our theoretical argument in Section 6. On the other hand, global entropy penalty increases correctness but comes at the cost of diversity. In tasks where diversity is more important, entropy bonus works well but in cases where diversity is less important, entropy penalty works well.

To quantify the importance of solution diversity for a given task, we propose the metric of *Solution Multiplicity*, defined as the average number of unique correct solutions per problem:

Solution Multiplicity$(\mathcal{X}) = \frac{1}{|\mathcal{X}|} \sum_{x \in \mathcal{X}} A(x)$, where A(x) is the number of solutions for problem x.

We measured this metric across four tasks (sampling 200 problems each) and correlated it with the change in Pass@8 performance from adding an entropy bonus. The results are presented below, with experimental details in Appendix C.4.

| Task | Knight and Knaves | Math | Countdown-3 | Countdown |
|---|---|---|---|---|
| **Solution Multiplicity** | 1.5 | 3.7 | 6.5 | 15.7 |
| **Entropy Effect** (for Pass@8) | -9.0% | -6.0% | +1.0% | +3.4% |

We conclude that when the number of unique solutions is larger, the benefit of increasing diversity outweighs the potential trade-off in single-solution correctness. Consequently, an entropy bonus is more favorable than an entropy penalty. This leads to our guiding principle for entropy control: for tasks characterized by high solution multiplicity, entropy bonus is beneficial but for a task with low solution multiplicity, entropy penalty is beneficial.

**Differential Control for Correct and Incorrect Trajectories.** The underlying mechanism of DS-GRPO is similar to a form of *differentiated entropy control*. An objective function representing this principle can be formulated as:

$$\mathcal{J}_{\text{DS-En}} = \mathcal{J}_{\text{GRPO}} - \eta_p \sum_{y:r(y)>0} \pi_\theta(y \mid x) \log \pi_\theta(\tau \mid x) + \eta_n \sum_{y:r(y)\leq 0} \pi_\theta(y \mid x) \log \pi_\theta(y \mid x).$$

By selectively increasing entropy only for positive samples, we attain the full diversity benefits of traditional entropy regularization, as we are only concerned with diversity among correct solutions. Concurrently, decreasing entropy for negative samples reinforces correctness. This targeted approach enables simultaneous gains in both correctness (Pass@1) and diversity (Pass@K), offering a more robust and principled method for model fine-tuning across different tasks.

> **Takeaway: Effect and Principle for Entropy Control**
>
> - **Inherent Trade-off:** A global entropy bonus enhances diversity at the cost of correctness, whereas an entropy penalty improves correctness but curtails diversity.

- **Task-Dependent Strategy:** For tasks with high complexity, an entropy bonus is more advantageous. The gains in diversity from exploration outweigh the potential reduction in single-solution accuracy.

- **Superiority of Differentiated Control:** DS-GRPO consistently outperforms both global entropy bonus and penalty strategies. This demonstrates controlling entropy differentially for correct and incorrect trajectories successfully captures the benefits of both approaches—enhancing diversity and reinforcing correctness simultaneously.

# 6 THEORETICAL ANALYSIS

We now theoretically establish the optimality of DS-GRPO. Our analysis proves its superiority over two baselines—Vanilla RL and RL with direct entropy maximization—and begins with the following formal definition:

**Definition 1** (Fine-tuned Policy). *We define differential smoothing policy, $\pi_{DS}$, parameterized by $\gamma_n$ and $\gamma_p$, as the solution to: $\pi_{DS}(\tau) = \arg\max_\pi \mathbb{E}_{\tau \sim \pi} r_{DS}(\tau) - \beta_{DS} \cdot \mathbb{D}_{KL}(\pi \| \pi_{base})$ where $r_{DS}$ is the modified reward defined in Eq. 2. As a baseline, we consider a policy that directly maximizes entropy, $\pi_{ent}$. It solves the same optimization problem, but replaces the reward $r_{DS}$ with $\widehat{r}(\tau) = r(\tau) - \gamma \log(\pi_{base}(\tau))$ for all trajectories $\tau$, and $\beta_{DS}$ with $\beta_{En}$. We further define policy for vanilla RL as the solution to the optimization problem in Eq. 1.*

To compare different methods, we introduce formal metrics for correctness and diversity.

**Definition 2** (Correctness and Correct-Solution Diversity). *For any policy $\pi$, we define its correctness as $C(\pi) = \sum_{\tau \in \mathcal{C}} \pi(\tau)$. We use the normalized variance on correct trajectories to measure diversity over correct solutions. Namely, we define $\sigma(\pi) = [\sum_{\tau \in \mathcal{C}} \pi(\tau)^2 - C(\pi)^2]/C(\pi)^2$.*

The normalized variance of our policy is smaller than that of vanilla RL, which, by our definition, corresponds to greater policy diversity. We now formally compare our method against the direct entropy maximization baseline.

> **Theoretical Guarantee for DS-GRPO**
>
> **Theorem 6.1.** *Assume the model have correct estimation for the reward of all trajectories. For any parameters $\gamma_{ent} \geq 0$ and $\beta_{ent} > 0$ used in Eq. 1 (for $\pi_{ent}$) that satisfy a proximity constraint $K_\rho(\pi_{ent}, \pi_{base}) \leq \kappa$, there exist parameters $\gamma_{DS} \geq 0$ and $\beta_{DS} > 0$ for $\pi_{DS}$ such that it also satisfies $K_\rho(\pi_{DS}, \pi_{base}) \leq \kappa$, and the following inequalities hold:*
>
> $$C(\pi_{DS}) \geq C(\pi_{ent}) \quad \text{and} \quad \sigma(\pi_{DS}) \geq \sigma(\pi_{ent}).$$
>
> *This result holds for $K_\rho(\pi, \pi_{base}) \in \{\mathbb{D}_{KL}(\pi \| \pi_{base}), \mathbb{D}_{KL}(\pi_{base} \| \pi), \mathbb{D}_{\chi^2}(\pi \| \pi_{base}), \mathbb{D}_{\chi^2}(\pi_{base} \| \pi)\}$.*

The KL-divergence constraint ($K_\rho(\cdot, \pi_{base}) \leq \kappa$) is a practical necessity and a standard assumption in prior work (Setlur et al., 2025). It prevents the fine-tuned policy from deviating excessively from the base model, thereby retaining pre-trained knowledge and avoiding catastrophic forgetting.

Our theoretical results show that our method surpasses both vanilla GRPO and direct entropy maximization in correctness and diversity, providing a formal justification for our strong empirical performance on Pass@1 and Pass@K metrics. A key insight from our analysis is the fundamental trade-off between these two objectives: increasing entropy enhances diversity at the potential cost of correctness, whereas an emphasis on correctness can harm diversity.

# 7 CONCLUSION

In this work, we conduct a formal, first-principles analysis of diversity collapse, from which we derive a novel method to enhance policy diversity. We empirically demonstrate that our method outperforms existing approaches and theoretically prove its optimality. Our analysis also clarifies the nuanced, task-dependent role of entropy in fine-tuning, leading to a principled control strategy that simultaneously improves both correctness (Pass@1) and diversity (Pass@K). A formal theoretical

analysis of our entropy principle and the nuanced effects of entropy it reveals is left as a promising direction for future research.

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

# Appendix

## Table of Contents

## A    ADDITIONAL RELATED WORK

**Mitigating diversity collapse in the RL of reasoning models.**    Reinforcement Learning with Verifiable Rewards (RLVR) has emerged as the dominant paradigm for enhancing LLM reasoning on tasks like mathematics and programming (Guo et al., 2025; Jaech et al., 2024). This process is often framed as "sharpening," where the model learns to place greater probability mass on high-quality sequences, thereby amortizing the high inference-time cost of generation (Huang et al., 2024; 2022; Wang et al., 2022; Bai et al., 2022b; Pang et al., 2023).

However, this self-improvement risks reducing creativity. Recent studies observe that RLVR often induces "diversity collapse," where the generation distribution becomes overly concentrated (Dang et al., 2025b; Yue et al., 2025). This collapse manifests empirically: despite higher pass@1 performance, models trained with RLVR (RLVR-trained models) often underperform their base model on pass@k for large $k$. This degradation limits test-time scaling and raises a fundamental question: does RLVR truly expand a model's reasoning capabilities, or does it merely sharpen the probability mass around solutions already present in the base distribution (Wu et al., 2025; Yue et al., 2025)?

To mitigate the problem of diversity collapse, a variety of approaches have been proposed from different perspectives. From the algorithm side, Yu et al. (2025) clip-higher strategy and the removal of KL divergence penalties in the GRPO of reasoning models, while He et al. (2025b) suggests adaptively using entropy as a form of regularization. Zhu et al. (2025) shows positive samples in RLVR sharpens the distribution around the sampled correct trajectories, whereas penalizing negative samples preserves diversity, motivating a higher weighting of negative samples in the training objective. In terms of reward design, several studies have proposed making rewards explicitly diversity-aware. Walder & Karkhanis (2025a); Chen et al. (2025c) suggest directly using pass@k metric as the reward. He et al. (2025a) introduces rank-based penalties within sampled groups to encourage diverse output, while Cui et al. (2025) incorporate entropy into advantage estimation to promote exploration. Other methods include interpolate the weights of the base model and the fine-tuned model (Dang et al., 2025b).

**Controlling distribution entropy in RLVR.**    The entropy of the policy distribution is a key internal indicator of a model's exploration capability(Cui et al., 2025; Cheng et al., 2025). Various methods have been proposed to maintain high entropy during training in order to encourage exploration, including clipping higher, adding entropy bonus (Yu et al., 2025; He et al., 2025b), or selectively training on critical high-entropy tokens (Wang et al., 2025). Other studies report that RLVR improves performance at the expense of reduced policy entropy (Cui et al., 2025), and that simply minimizing entropy can effectively improve pass@1 accuracy(Agarwal et al., 2025; Gao et al., 2025). Xingjin Wang (2025) further propose an entropy scheduling approach that maintains high entropy in the early stage to encourage exploration and reduces entropy later to improve final performance. In contrast to prior approaches, we treat correct and incorrect samples separately: bonusing entropy for correct samples and penalizing entropy for incorrect ones. We demonstrate the superiority of this design both theoretically and empirically.

**Sharpening in RL prior to language model.**    Diversity collapse is not unique to language models; it has been extensively observed in broader reinforcement learning settings (Hong et al., 2018; Haarnoja, 2018; Schulman et al., 2017; Chi et al., 2025; Jabri, 2021) . For instance, in traditional RL domains such as 2D Gridworlds or Atari 2600 (Hong et al., 2018), agents exhibit a strong bias toward learning policies and actions that they are initially more confident in or that are easier to access during early exploration.

The fundamental cause of sharpening in traditional RL aligns with what we observe in LLMs. Specifically, states or actions that have a higher initial probability of being visited are explored more frequently. Consequently, even if a certain state yields a higher reward, it is less likely to be discovered and reinforced if its initial reachability is low. This "rich-get-richer" dynamic in exploration distribution drives the sharpening effect in both domains.

We specifically focus on RL with Verifiable Rewards (RLVR) for LLMs because the sharpening effect here is particularly acute compared to traditional settings. Unlike many control and robotics tasks where dense process rewards are often available along the trajectory, RL for LLMs typically relies on outcome-only rewards. Optimization focuses almost entirely on a sparse signal at the

final token. As shown in recent studies (Dang et al., 2025a; Kirk et al., 2023), this setup strongly amplifies mode-seeking behavior (Yin et al., 2025; Ladosz et al., 2022), making diversity collapse a more critical issue in LLMs than in environments with denser feedback signals.

# B  PROOF OF THEOREMS

## B.1  PROOF OF PROPOSITION B.1 AND PROPOSITION B.2

**Lemma 1.** *The solution to the KL-regularized optimization problem:*

$$\pi^*_{\beta_{ent}} = \arg\max_{\pi} \left\{ \mathbb{E}_{\tau \sim \pi} r(\tau) - \beta_{ent} \cdot \mathbb{D}_{KL}(\pi||\pi_{base}) \right\}$$

*has the following closed-form expression for a trajectory $\tau$:*

$$\pi^*_{\beta_{ent}(\tau)} = \frac{\left[ \prod_{h=1}^{H} \pi_{base,h}(a_h \mid s_h) \right] \exp\left( \frac{1}{\beta_{ent}} r(\tau) \right)}{\sum_{\tau' : s'_1 = s} \left[ \prod_{h=1}^{H} \pi_{base,h}(a'_h \mid s'_h) \right] \exp\left( \frac{1}{\beta_{ent}} r(\tau') \right)},$$

*where the summation in the denominator is over all valid trajectories $\tau'$ starting from the initial state $s$.*

*Proof.* The optimization problem can be written as finding a probability distribution $\pi(\tau)$ over trajectories that solves:

$$\max_{\pi} \sum_{\tau} \pi(\tau) r(\tau) - \beta_{ent} \sum_{\tau} \pi(\tau) \ln\left( \frac{\pi(\tau)}{\pi_{base}(\tau)} \right), \quad \text{subject to} \quad \sum_{\tau} \pi(\tau) = 1.$$

We introduce a Lagrange multiplier $\mu$ for the probability constraint and form the Lagrangian $\mathcal{L}(\pi, \mu)$:

$$\mathcal{L}(\pi, \mu) = \sum_{\tau} \pi(\tau) r(\tau) - \beta_{ent} \sum_{\tau} \pi(\tau) \left[ \ln(\pi(\tau)) - \ln(\pi_{base}(\tau)) \right] - \mu \left( \sum_{\tau} \pi(\tau) - 1 \right).$$

To find the optimal policy, we take the partial derivative of $\mathcal{L}$ with respect to $\pi(\tau)$ and set it to zero:

$$\frac{\partial \mathcal{L}}{\partial \pi(\tau)} = r(\tau) - \beta_{ent} \left( \ln\left( \frac{\pi(\tau)}{\pi_{base}(\tau)} \right) + 1 \right) - \mu = 0.$$

Solving for $\pi(\tau)$, we obtain:

$$\ln\left( \frac{\pi(\tau)}{\pi_{base}(\tau)} \right) = \frac{r(\tau)}{\beta_{ent}} - 1 - \frac{\mu}{\beta_{ent}}$$

$$\implies \pi(\tau) = \pi_{base}(\tau) \exp\left( \frac{r(\tau)}{\beta_{ent}} - 1 - \frac{\mu}{\beta_{ent}} \right) = \pi_{base}(\tau) \exp\left( \frac{r(\tau)}{\beta_{ent}} \right) \exp\left( -1 - \frac{\mu}{\beta_{ent}} \right).$$

The term $\exp(-1 - \mu/\beta_{ent})$ is a constant determined by the normalization constraint $\sum_{\tau'} \pi(\tau') = 1$. Let the partition function be $\mathcal{Z} = \sum_{\tau'} \pi_{base}(\tau') \exp\left( \frac{r(\tau')}{\beta_{ent}} \right)$. The normalization constant must be $1/\mathcal{Z}$, which gives the solution:

$$\pi^*_{\beta_{ent}}(\tau) = \frac{\pi_{base}(\tau) \exp\left( \frac{1}{\beta_{ent}} r(\tau) \right)}{\mathcal{Z}} = \frac{\pi_{base}(\tau) \exp\left( \frac{1}{\beta_{ent}} r(\tau) \right)}{\sum_{\tau'} \pi_{base}(\tau') \exp\left( \frac{1}{\beta_{ent}} r(\tau') \right)}.$$

By substituting the definitions $\pi_{base}(\tau) = \prod_{h=1}^{H} \pi_{base,h}(a_h \mid s_h)$, we arrive at the expression stated in the lemma. This completes the proof. $\square$

**Proposition B.1** (Selection Bias). *The probability that a correct trajectory's likelihood increases is monotonically related to its initial probability under the base model. Formally, for any two correct trajectories $\tau_1, \tau_2$ and $\beta_{ent} > 0$, we have*

$$\pi_{base}(\tau_1) \geq \pi_{base}(\tau_2) \implies \mathbb{P}(\pi^*_{van}(\tau_1) > \pi_{base}(\tau_1)) \geq \mathbb{P}(\pi^*_{van}(\tau_2) > \pi_{base}(\tau_2)).$$

**Proposition B.2** (Reinforcement bias). *The magnitude of probability gain for a given trajectory is directly proportional to its probability under the base policy. Formally, if the reward update mechanism has access to the complete set of correct trajectories ($r(\tau) = 1$ for all correct trajectories), then for any trajectory $\tau$ and $\beta_{ent} > 0$, we have*

$$\pi^*_{van}(\tau) - \pi_{base}(\tau) \propto \pi_{base}(\tau).$$

*Proof.* **Part 1: Monotonicity of Likelihood Improvement.** Under the specified reward mechanism, a correct trajectory $\tau$ receives a positive reward if it is not missed in all $N$ independent samples. This occurs with probability $1 - (1 - \pi_{\text{base}}(\tau))^N$. A positive reward ensures that the likelihood of the trajectory increases after fine-tuning. Thus, the probability of improvement is:

$$\mathbb{P}(\pi_{\text{van}}^*(\tau) > \pi_{\text{base}}(\tau)) = 1 - (1 - \pi_{\text{base}}(\tau))^N.$$

This function is monotonically increasing with respect to $\pi_{\text{base}}(\tau)$ for $\pi_{\text{base}}(\tau) \in [0, 1]$. Therefore, if $\pi_{\text{base}}(\tau_1) \geq \pi_{\text{base}}(\tau_2)$, the first claim holds.

**Part 2: Proportionality of Probability Gain.** From the closed-form solution for the optimal policy (as derived in Lemma 1), we have $\pi_{\beta_{\text{ent}}}^*(\tau) = \pi_{\text{base}}(\tau) \exp(r(\tau)/\beta_{\text{ent}})/\mathcal{Z}$, where $\mathcal{Z}$ is the partition function. The change in probability is:

$$\pi_{\beta_{\text{ent}}}^*(\tau) - \pi_{\text{base}}(\tau) = \frac{\pi_{\text{base}}(\tau) \exp(r(\tau)/\beta_{\text{ent}})}{\mathcal{Z}} - \pi_{\text{base}}(\tau) = \pi_{\text{base}}(\tau) \left[ \frac{\exp(r(\tau)/\beta_{\text{ent}}) - \mathcal{Z}}{\mathcal{Z}} \right].$$

The partition function $\mathcal{Z} = \sum_{\tau'} \pi_{\text{base}}(\tau') \exp(r(\tau')/\beta_{\text{ent}})$ is a constant for a given policy $\pi_{\text{base}}$ and reward function $r$. The term in the brackets is therefore constant for all trajectories $\tau$ that share the same reward value (e.g., all correct trajectories). Consequently, the probability gain is directly proportional to the initial probability $\pi_{\text{base}}(\tau)$. $\qquad\square$

## B.2 Explaining Sharpening in Practical RL with Theoretical Model.

**Verifier-based RL.** In practice, verifier-based RL algorithms typically maintain two policies: a sampling policy and a learned policy. The training process is iterative: the model uses a fixed sampling policy to generate trajectories for $t$ iterations, updating the learned policy at each step. After $t$ iterations, the sampling policy is updated to match the current learned policy. Our theoretical model abstracts the learning dynamics within these $t$ iterations where the sampling policy is held fixed. We demonstrate that within each such phase, the distribution over correct answers is sharpened.

Although our formal analysis focuses on the specific phase where the sampling policy is fixed, the full training process can be viewed as a cumulative composition of these phases. Since the model tends to sharpen the answer distribution within each iteration window (as shown in our theory), the aggregate effect over the entire learning process inevitably leads to a globally sharpened distribution over answers.

**RL with other forms of reward function.** While our work primarily addresses RL with verifiers, our insights extend to settings with learned reward models, such as RLHF. Specifically, our theory highlights that since the model samples answers from a base distribution, high-probability correct answers are sampled—and thus reinforced—more frequently. A similar mechanism applies to RLHF. During the training of the reward model (or the policy based on it), the system often relies on samples from the base model. Between two equally favorable responses, the one with a higher initial sampling probability is likely to be exposed more often, leading the model to preferentially favor and amplify it. Thus, the "rich get richer" dynamic contributes to sharpening in these settings as well.

## B.3 Proof of Theorem 6.1

**Lemma 2.** *The solution to the KL-regularized optimization problem with an entropy-based reward modification:*

$$\pi_{\beta_{ent}, \gamma_{ent}}^* = \arg \max_\pi \left\{ \mathbb{E}_{\tau \sim \pi} \left[ r(\tau) - \gamma_{ent} \log(\pi_{base}(\tau)) \right] - \beta_{ent} \cdot \mathbb{D}_{KL}(\pi \| \pi_{base}) \right\}$$

*is given by:*

$$\pi_{\beta_{ent}, \gamma_{ent}}^*(\tau) = \frac{[\pi_{base}(\tau)]^{1 - \frac{\gamma_{ent}}{\beta_{ent}}} \exp \left( \frac{1}{\beta_{ent}} r(\tau) \right)}{\sum_{\tau'} [\pi_{base}(\tau')]^{1 - \frac{\gamma_{ent}}{\beta_{ent}}} \exp \left( \frac{1}{\beta_{ent}} r(\tau') \right)},$$

*where the summation in the denominator is over all valid trajectories $\tau'$.*

*Proof.* The objective function can be expanded as:

$$\max_{\pi} \sum_{\tau} \pi(\tau) \left( r(\tau) - \gamma_{\text{ent}} \log(\pi_{\text{base}}(\tau)) \right) - \beta_{\text{ent}} \sum_{\tau} \pi(\tau) \ln \left( \frac{\pi(\tau)}{\pi_{\text{base}}(\tau)} \right),$$

subject to the constraint $\sum_{\tau} \pi(\tau) = 1$. We form the Lagrangian $\mathcal{L}(\pi, \mu)$:

$$\mathcal{L}(\pi, \mu)$$

$$= \sum_{\tau} \pi(\tau) \left( r(\tau) - \gamma_{\text{ent}} \log(\pi_{\text{base}}(\tau)) \right) - \beta_{\text{ent}} \sum_{\tau} \pi(\tau) \left( \ln(\pi(\tau)) - \ln(\pi_{\text{base}}(\tau)) \right) - \mu \left( \sum_{\tau} \pi(\tau) - 1 \right).$$

Setting the partial derivative with respect to $\pi(\tau)$ to zero yields:

$$\frac{\partial \mathcal{L}}{\partial \pi(\tau)} = r(\tau) - \gamma_{\text{ent}} \log(\pi_{\text{base}}(\tau)) - \beta_{\text{ent}} \left( \ln \left( \frac{\pi(\tau)}{\pi_{\text{base}}(\tau)} \right) + 1 \right) - \mu = 0.$$

Solving for $\pi(\tau)$:

$$\ln \left( \frac{\pi(\tau)}{\pi_{\text{base}}(\tau)} \right) = \frac{r(\tau)}{\beta_{\text{ent}}} - \frac{\gamma_{\text{ent}}}{\beta_{\text{ent}}} \log(\pi_{\text{base}}(\tau)) - 1 - \frac{\mu}{\beta_{\text{ent}}}$$

$$\implies \pi(\tau) = \pi_{\text{base}}(\tau) \exp \left( \frac{r(\tau)}{\beta_{\text{ent}}} - \frac{\gamma_{\text{ent}}}{\beta_{\text{ent}}} \log(\pi_{\text{base}}(\tau)) - 1 - \frac{\mu}{\beta_{\text{ent}}} \right)$$

$$= \pi_{\text{base}}(\tau) \cdot (\exp(\log(\pi_{\text{base}}(\tau))))^{-\frac{\gamma_{\text{ent}}}{\beta_{\text{ent}}}} \cdot \exp \left( \frac{r(\tau)}{\beta_{\text{ent}}} \right) \cdot \exp \left( -1 - \frac{\mu}{\beta_{\text{ent}}} \right)$$

$$= [\pi_{\text{base}}(\tau)]^{1 - \frac{\gamma_{\text{ent}}}{\beta_{\text{ent}}}} \exp \left( \frac{r(\tau)}{\beta_{\text{ent}}} \right) \exp \left( -1 - \frac{\mu}{\beta_{\text{ent}}} \right).$$

The term $\exp(-1 - \mu/\beta_{\text{ent}})$ is a normalization constant. By enforcing the constraint $\sum_{\tau'} \pi(\tau') = 1$, we find that this constant is the reciprocal of the partition function $\mathcal{Z} = \sum_{\tau'} [\pi_{\text{base}}(\tau')]^{1 - \frac{\gamma_{\text{ent}}}{\beta_{\text{ent}}}} \exp(r(\tau')/\beta_{\text{ent}})$. This gives the final solution stated in the lemma. $\qquad \square$

**Lemma 3.** *Consider the reward function $r_{DS}(\tau)$ which modifies the reward based on trajectory correctness, defined by a set of correct trajectories $\mathcal{C}$:*

$$r_{DS}(\tau) = \begin{cases} r(\tau) - \gamma_{DS} \log(\pi_{base}(\tau)) & \text{if } \tau \in \mathcal{C} \\ r(\tau) & \text{if } \tau \notin \mathcal{C} \end{cases}$$

*The solution to the KL-regularized optimization problem $\pi_{DS} = \arg\max_{\pi} \{ \mathbb{E}_{\tau \sim \pi}[r_{DS}(\tau)] - \beta_{DS} \cdot \mathbb{D}_{KL}(\pi || \pi_{base}) \}$ is given by:*

$$\pi_{DS}(\tau) = \frac{1}{\mathcal{Z}} \times \begin{cases} [\pi_{base}(\tau)]^{1 - \frac{\gamma_{DS}}{\beta_{DS}}} \exp \left( \frac{1}{\beta_{DS}} r(\tau) \right) & \text{if } \tau \in \mathcal{C} \\ [\pi_{base}(\tau)] \cdot \exp \left( \frac{1}{\beta_{DS}} r(\tau) \right) & \text{if } \tau \notin \mathcal{C} \end{cases}$$

*where $\mathcal{Z}$ is the partition function ensuring normalization.*

*Proof.* The objective function is maximized subject to $\sum_{\tau} \pi(\tau) = 1$. The Lagrangian is:

$$\mathcal{L}(\pi, \mu) = \sum_{\tau \in \mathcal{C}} \pi(\tau) \left( r(\tau) - \gamma_{\text{DS}} \log(\pi_{\text{base}}(\tau)) \right) + \sum_{\tau \notin \mathcal{C}} \pi(\tau) r(\tau)$$

$$- \beta_{\text{DS}} \sum_{\tau} \pi(\tau) \left( \ln(\pi(\tau)) - \ln(\pi_{\text{base}}(\tau)) \right) - \mu \left( \sum_{\tau} \pi(\tau) - 1 \right).$$

We take the partial derivative with respect to $\pi(\tau)$ for each case and set it to zero.

For a correct trajectory, $\tau \in \mathcal{C}$:

$$\frac{\partial \mathcal{L}}{\partial \pi(\tau)} = r(\tau) - \gamma_{\text{DS}} \log(\pi_{\text{base}}(\tau)) - \beta_{\text{DS}} \left( \ln \left( \frac{\pi(\tau)}{\pi_{\text{base}}(\tau)} \right) + 1 \right) - \mu = 0.$$

Solving for $\pi(\tau)$ yields: $\pi(\tau) \propto [\pi_{\text{base}}(\tau)]^{1-\frac{\gamma_{\text{DS}}}{\beta_{\text{DS}}}} \exp\left(\frac{r(\tau)}{\beta_{\text{DS}}}\right)$.

For an incorrect trajectory, $\tau \notin \mathcal{C}$:

$$\frac{\partial \mathcal{L}}{\partial \pi(\tau)} = r(\tau) - \beta_{\text{DS}} \cdot \left(\ln\left(\frac{\pi(\tau)}{\pi_{\text{base}}(\tau)}\right) + 1\right) - \mu = 0.$$

Solving for $\pi(\tau)$ yields: $\pi(\tau) \propto [\pi_{\text{base}}(\tau)] \cdot \exp\left(\frac{r(\tau)}{\beta_{\text{ent}}}\right)$.

Combining these results, the unnormalized solution $\tilde{\pi}(\tau)$ is:

$$\tilde{\pi}(\tau) = \begin{cases} [\pi_{\text{base}}(\tau)]^{1-\frac{\gamma_{\text{DS}}}{\beta_{\text{ent}}}} \exp\left(\frac{1}{\beta_{\text{ent}}} r(\tau)\right) & \text{if } \tau \in \mathcal{C} \\ [\pi_{\text{base}}(\tau)] \cdot \exp\left(\frac{1}{\beta_{\text{ent}}} r(\tau)\right) & \text{if } \tau \notin \mathcal{C} \end{cases}$$

The final solution $\pi_{\text{DS}}$ is obtained by normalizing $\tilde{\pi}(\tau)$ with the partition function $\mathcal{Z} = \sum_{\tau'} \tilde{\pi}(\tau')$, which gives the expression stated in the lemma. $\qquad\square$

**Lemma 4** (Correctness under Reverse KL Constraint). *When $K_\rho(\pi, \pi_{base}) = \mathbb{D}_{KL}(\pi_{base}\|\pi)$, for any $\gamma_{ent} \geq 0, \beta_{ent} > 0$ such that $\mathbb{D}_{KL}(\pi_{base}\|\pi_{ent}) \leq \kappa$, there exist $\gamma_{DS} \geq 0$ and $\beta_{DS} > 0$ such that $\mathbb{D}_{KL}(\pi_{base}\|\pi_{DS}) \leq \kappa$ and $C(\pi_{DS}) \geq C(\pi_{ent})$.*

*Proof.* To simplify the notation, we define the following sums over trajectory probabilities, where $\mathcal{C}$ is the set of correct trajectories:

$$b_x = \sum_{\tau \in \mathcal{C}} [\pi_{\text{base}}(\tau)]^{1-x}, \quad B_x = \sum_{\tau} [\pi_{\text{base}}(\tau)]^{1-x}, \quad p_c = \sum_{\tau \in \mathcal{C}} \pi_{\text{base}}(\tau).$$

The proof proceeds in three steps: we first find a functional relationship between correctness $C$ and the KL divergence for each policy, and then compare them.

**Step 1: Analyze the Entropy-Maximization Policy ($\pi_{\text{ent}}$).** The correctness is the total probability mass on correct trajectories:

$$C(\pi_{\text{ent}}) = \frac{\sum_{\tau \in \mathcal{C}} [\pi_{\text{base}}(\tau)]^{1-\frac{\gamma_{\text{ent}}}{\beta_{\text{ent}}}} \exp(1/\beta_{\text{ent}})}{\sum_{\tau \in \mathcal{C}} [\pi_{\text{base}}(\tau)]^{1-\frac{\gamma_{\text{ent}}}{\beta_{\text{ent}}}} \exp(1/\beta_{\text{ent}}) + \sum_{\tau \notin \mathcal{C}} [\pi_{\text{base}}(\tau)]^{1-\frac{\gamma_{\text{ent}}}{\beta_{\text{ent}}}}} = \frac{b_{\frac{\gamma_{\text{ent}}}{\beta_{\text{ent}}}} e^{1/\beta_{\text{ent}}}}{b_{\frac{\gamma_{\text{ent}}}{\beta_{\text{ent}}}} e^{1/\beta_{\text{ent}}} + (B_{\frac{\gamma_{\text{ent}}}{\beta_{\text{ent}}}} - b_{\frac{\gamma_{\text{ent}}}{\beta_{\text{ent}}}})}.$$

Solving for $e^{1/\beta_{\text{ent}}}$ gives: $e^{1/\beta_{\text{ent}}} = \frac{B_{\frac{\gamma_{\text{ent}}}{\beta_{\text{ent}}}} - b_{\frac{\gamma_{\text{ent}}}{\beta_{\text{ent}}}}}{b_{\frac{\gamma_{\text{ent}}}{\beta_{\text{ent}}}}} \left(\frac{C(\pi_{\text{ent}})}{1-C(\pi_{\text{ent}})}\right)$. The reverse KL divergence is $\mathbb{D}_{\text{KL}}(\pi_{\text{base}}\|\pi_{\text{ent}}) = \sum_\tau \pi_{\text{base}}(\tau) \ln(\pi_{\text{base}}(\tau)/\pi_{\text{ent}}(\tau))$. Substituting the policy definition:

$$\mathbb{D}_{\text{KL}}(\pi_{\text{base}}\|\pi_{\text{ent}}) = \frac{\gamma_{\text{ent}}}{\beta_{\text{ent}}} \sum_\tau \pi_{\text{base}} \ln \pi_{\text{base}} - \frac{1}{\beta_{\text{ent}}} \sum_\tau \pi_{\text{base}} \cdot r(\tau) + \ln\left(b_{\frac{\gamma_{\text{ent}}}{\beta_{\text{ent}}}} e^{1/\beta_{\text{ent}}} + B_{\frac{\gamma_{\text{ent}}}{\beta_{\text{ent}}}} - b_{\frac{\gamma_{\text{ent}}}{\beta_{\text{ent}}}}\right)$$

$$= \frac{\gamma_{\text{ent}}}{\beta_{\text{ent}}} \sum_\tau \pi_{\text{base}} \ln \pi_{\text{base}} - \frac{p_c}{\beta_{\text{ent}}} + \ln\left(\frac{b_{\frac{\gamma_{\text{ent}}}{\beta_{\text{ent}}}} e^{1/\beta_{\text{ent}}}}{C(\pi_{\text{ent}})}\right).$$

Substituting the expression for $1/\beta_{\text{ent}}$ leads to a relationship between divergence and correctness:

$$\mathbb{D}_{\text{KL}}(\pi_{\text{base}}\|\pi_{\text{ent}}) = \frac{\gamma_{\text{ent}}}{\beta_{\text{ent}}} \sum_\tau \pi_{\text{base}} \ln \pi_{\text{base}} + p_c \ln b_{\frac{\gamma_{\text{ent}}}{\beta_{\text{ent}}}} + (1-p_c) \ln(B_{\frac{\gamma_{\text{ent}}}{\beta_{\text{ent}}}} - b_{\frac{\gamma_{\text{ent}}}{\beta_{\text{ent}}}})$$

$$- [p_c \ln C(\pi_{\text{ent}}) + (1-p_c) \ln(1 - C(\pi_{\text{ent}}))].$$

**Step 2: Analyze Differential Policy ($\pi_{\text{DS}}$).** Similarly, the correctness is:

$$C(\pi_{\text{DS}}) = \frac{b_{\frac{\gamma_{\text{DS}}}{\beta_{\text{DS}}}} e^{1/\beta_{\text{DS}}}}{b_{\frac{\gamma_{\text{DS}}}{\beta_{\text{DS}}}} e^{1/\beta_{\text{DS}}} + (1-p_c)}.$$

The reverse KL divergence, after a similar derivation, is:

$$\mathbb{D}_{\text{KL}}(\pi_{\text{base}}\|\pi_{\text{DS}}) = \frac{\gamma_{\text{DS}}}{\beta_{\text{DS}}} \sum_{\tau \in \mathcal{C}} \pi_{\text{base}} \ln \pi_{\text{base}} + p_c \ln b_{\frac{\gamma_{\text{DS}}}{\beta_{\text{DS}}}} + (1-p_c) \ln(1-p_c)$$

$$- \left[ p_c \ln C(\pi_{\mathrm{DS}}) + (1 - p_c) \ln(1 - C(\pi_{\mathrm{DS}})) \right]. \tag{5}$$

**Step 3: Compare the Policies.** Our goal is to show that for any $C(\pi_{\mathrm{ent}})$, we can choose parameters for our method to achieve $C(\pi_{\mathrm{DS}}) = C(\pi_{\mathrm{ent}})$ with a smaller or equal KL divergence. Let's choose $\gamma_{\mathrm{DS}} = \frac{\beta_{\mathrm{DS}}}{\beta_{\mathrm{ent}}} \gamma_{\mathrm{ent}}$, and set $C(\pi_{\mathrm{DS}}) = C(\pi_{\mathrm{ent}}) = C$. Equivalently, we assume that

$$\frac{\gamma_{\mathrm{DS}}}{\beta_{\mathrm{DS}}} = \frac{\gamma_{\mathrm{ent}}}{\beta_{\mathrm{ent}}} = \widetilde{\gamma}.$$

Then the KL divergence for our method becomes:

$$\mathbb{D}_{\mathrm{KL}}(\pi_{\mathrm{base}} \| \pi_{\mathrm{DS}}) = \widetilde{\gamma} \sum_{\tau \in \mathcal{C}} \pi_{\mathrm{base}} \ln \pi_{\mathrm{base}} + p_c \ln b_{\gamma_{\mathrm{ent}}} + (1 - p_c) \ln(1 - p_c) - H(C_{\mathrm{ent}}),$$

where $H(C) = -[p_c \ln C + (1 - p_c) \ln(1 - C)]$. For the entropy method, the KL is:

$$\mathbb{D}_{\mathrm{KL}}(\pi_{\mathrm{base}} \| \pi_{\mathrm{ent}}) = \widetilde{\gamma} \sum_{\tau \in \mathcal{C}} \pi_{\mathrm{base}} \ln \pi_{\mathrm{base}} + \widetilde{\gamma} \sum_{\tau \notin \mathcal{C}} \pi_{\mathrm{base}} \ln \pi_{\mathrm{base}} + p_c \ln b_{\widetilde{\gamma}} + (1 - p_c) \ln(B_{\widetilde{\gamma}} - b_{\widetilde{\gamma}}) - H(C_{\mathrm{ent}}).$$

The difference is $\mathbb{D}_{\mathrm{KL}}(\pi_{\mathrm{base}} \| \pi_{\mathrm{ent}}) - \mathbb{D}_{\mathrm{KL}}(\text{ours}) = \widetilde{\gamma} \sum_{\tau \notin \mathcal{C}} \pi_{\mathrm{base}} \ln \pi_{\mathrm{base}} + (1 - p_c) \ln(B_{\widetilde{\gamma}} - b_{\widetilde{\gamma}}) - (1 - p_c) \ln(1 - p_c)$. By Jensen's inequality on the concave function $\ln(\cdot)$:

$$\sum_{\tau \notin \mathcal{C}} \frac{\pi_{\mathrm{base}}(\tau)}{1 - p_c} \ln \left( [\pi_{\mathrm{base}}(\tau)]^{-\widetilde{\gamma}} \right) \leq \ln \left( \sum_{\tau \notin \mathcal{C}} \frac{\pi_{\mathrm{base}}(\tau)}{1 - p_c} [\pi_{\mathrm{base}}(\tau)]^{-\widetilde{\gamma}} \right) = \ln \left( \frac{B_{\widetilde{\gamma}} - b_{\widetilde{\gamma}}}{1 - p_c} \right)$$

Multiplying by $-(1 - p_c)$ gives:

$$\widetilde{\gamma} \sum_{\tau \notin \mathcal{C}} \pi_{\mathrm{base}}(\tau) \ln(\pi_{\mathrm{base}}(\tau)) \geq -(1 - p_c) \ln \left( \frac{B_{\widetilde{\gamma}} - b_{\widetilde{\gamma}}}{1 - p_c} \right) = -(1 - p_c) \left[ \ln(B_{\widetilde{\gamma}} - b_{\widetilde{\gamma}}) - \ln(1 - p_c) \right].$$

Therefore, the difference is non-negative: $\mathbb{D}_{\mathrm{KL}}(\pi_{\mathrm{base}} \| \pi_{\mathrm{ent}}) - \mathbb{D}_{\mathrm{KL}}(\pi_{\mathrm{base}} \| \pi_{\mathrm{DS}}) \geq 0$. This means that for any given correctness level $C$, our method (with $\gamma_{\mathrm{DS}} = \gamma_{\mathrm{ent}} \cdot \frac{\beta_{\mathrm{DS}}}{\beta_{\mathrm{ent}}}$) can achieve it with a lower or equal KL-divergence cost. Thus, if both methods is constrained by $\mathbb{D}_{\mathrm{KL}}(\pi_{\mathrm{base}} \| \pi_{\mathrm{ent}}) \leq \kappa$ and $\mathbb{D}_{\mathrm{KL}}(\pi_{\mathrm{base}} \| \pi_{\mathrm{DS}}) \leq \kappa$, our method can achieve a correctness $C(\pi_{\mathrm{DS}}) \geq C(\pi_{\mathrm{ent}})$. $\qquad \square$

**Lemma 5** (Correctness under Forward KL Constraint). *When $K_\rho(\pi, \pi_{base}) = \mathbb{D}_{KL}(\pi \| \pi_{base})$, for any $\gamma_{ent} \geq 0, \beta_{ent} > 0$ such that $\mathbb{D}_{KL}(\pi_{ent} \| \pi_{base}) \leq \kappa$, there exist $\gamma_{DS} \geq 0$ and $\beta_{DS} > 0$ such that $\mathbb{D}_{KL}(\pi_{DS} \| \pi_{base}) \leq \kappa$ and $C(\pi_{DS}) \geq C(\pi_{ent})$.*

*Proof.* The proof proceeds in three steps: we first find a functional relationship between correctness $C$ and the KL divergence for each policy, and then compare them.

**Step 1: Analyze the Entropy-Maximization Policy ($\pi_{\mathrm{ent}}$).** The correctness of $\pi_{ent}$ is the total probability mass on correct trajectories:

$$C(\pi_{\mathrm{ent}}) = \frac{\sum_{\tau \in \mathcal{C}} [\pi_{\mathrm{base}}(\tau)]^{1 - \frac{\gamma_{\mathrm{ent}}}{\beta_{\mathrm{ent}}}} \exp(1/\beta_{\mathrm{ent}})}{\sum_{\tau \in \mathcal{T}} [\pi_{\mathrm{base}}(\tau)]^{1 - \frac{\gamma_{\mathrm{ent}}}{\beta_{\mathrm{ent}}}} \exp(r(\tau)/\beta_{\mathrm{ent}})} = \frac{b_{\frac{\gamma_{\mathrm{ent}}}{\beta_{\mathrm{ent}}}} e^{1/\beta_{\mathrm{ent}}}}{b_{\frac{\gamma_{\mathrm{ent}}}{\beta_{\mathrm{ent}}}} e^{1/\beta_{\mathrm{ent}}} + \left( B_{\frac{\gamma_{\mathrm{ent}}}{\beta_{\mathrm{ent}}}} - b_{\frac{\gamma_{\mathrm{ent}}}{\beta_{\mathrm{ent}}}} \right)}. \tag{6}$$

Solving for $e^{1/\beta_{\mathrm{ent}}}$ yields: $e^{1/\beta_{\mathrm{ent}}} = \frac{B_{\frac{\gamma_{\mathrm{ent}}}{\beta_{\mathrm{ent}}}} - b_{\frac{\gamma_{\mathrm{ent}}}{\beta_{\mathrm{ent}}}}}{b_{\frac{\gamma_{\mathrm{ent}}}{\beta_{\mathrm{ent}}}}} \left( \frac{C(\pi_{\mathrm{ent}})}{1 - C(\pi_{\mathrm{ent}})} \right).$

The reverse KL divergence $\mathbb{D}_{\mathrm{KL}}(\pi_{\mathrm{ent}} \| \pi_{\mathrm{base}})$ can be expressed as a function of $C(\pi_{\mathrm{ent}})$. Following the derivation previously, we arrive at:

$$\mathbb{D}_{\mathrm{KL}}(\pi_{\mathrm{ent}} \| \pi_{\mathrm{base}}) = -\frac{\frac{\gamma_{\mathrm{ent}}}{\beta_{\mathrm{ent}}}}{b_{\frac{\gamma_{\mathrm{ent}}}{\beta_{\mathrm{ent}}}}} \sum_{\tau \in \mathcal{C}} \pi_{\mathrm{base}}(\tau)^{1 - \frac{\gamma_{\mathrm{ent}}}{\beta_{\mathrm{ent}}}} \ln(\pi_{\mathrm{base}}(\tau)) \cdot C(\pi_{\mathrm{ent}})$$

$$- \frac{\frac{\gamma_{\mathrm{ent}}}{\beta_{\mathrm{ent}}}}{B_{\frac{\gamma_{\mathrm{ent}}}{\beta_{\mathrm{ent}}}} - b_{\frac{\gamma_{\mathrm{ent}}}{\beta_{\mathrm{ent}}}}} \sum_{\tau \notin \mathcal{C}} \pi_{\mathrm{base}}(\tau)^{1 - \frac{\gamma_{\mathrm{ent}}}{\beta_{\mathrm{ent}}}} \ln(\pi_{\mathrm{base}}(\tau)) \cdot (1 - C(\pi_{\mathrm{ent}})) \tag{7}$$

$$+ (1 - C(\pi_{\mathrm{ent}})) \ln \left( \frac{b_{\frac{\gamma_{\mathrm{ent}}}{\beta_{\mathrm{ent}}}}}{B_{\frac{\gamma_{\mathrm{ent}}}{\beta_{\mathrm{ent}}}} - b_{\frac{\gamma_{\mathrm{ent}}}{\beta_{\mathrm{ent}}}}} \right) - \ln b_{\frac{\gamma_{\mathrm{ent}}}{\beta_{\mathrm{ent}}}} + H(C(\pi_{\mathrm{ent}})),$$

where $H(C) = C \ln C + (1 - C) \ln(1 - C)$ is the binary entropy function.

**Step 2: Analyze Differential Policy ($\pi_{\mathbf{DS}}$).** Similarly, the correctness for our policy is given by:

$$C(\pi_{\text{DS}}) = \frac{b_{\frac{\gamma_{\text{DS}}}{\beta_{\text{DS}}}} e^{1/\beta_{\text{DS}}}}{b_{\frac{\gamma_{\text{DS}}}{\beta_{\text{DS}}}} e^{1/\beta_{\text{DS}}} + (1 - p_c)}. \tag{8}$$

The corresponding reverse KL divergence as a function of $C(\pi_{\text{DS}})$ is:

$$\mathbb{D}_{\text{KL}}(\pi_{\text{DS}}\|\pi_{\text{base}}) = -\frac{\frac{\gamma_{\text{DS}}}{\beta_{\text{DS}}}}{b_{\frac{\gamma_{\text{DS}}}{\beta_{\text{DS}}}}} \sum_{\tau \in \mathcal{C}} \pi_{\text{base}}(\tau)^{1 - \frac{\gamma_{\text{DS}}}{\beta_{\text{DS}}}} \ln(\pi_{\text{base}}(\tau)) \cdot C(\pi_{\text{DS}})$$

$$+ (1 - C(\pi_{\text{DS}})) \ln\left(\frac{b_{\frac{\gamma_{\text{DS}}}{\beta_{\text{DS}}}}}{1 - p_c}\right) - \ln b_{\frac{\gamma_{\text{DS}}}{\beta_{\text{DS}}}} + H(C(\pi_{\text{DS}})). \tag{9}$$

**Step 3: Compare the Policies.** Our goal is to show that for any $C(\pi_{\text{ent}})$, we can choose parameters for our method to achieve $C(\pi_{\text{DS}}) = C(\pi_{\text{ent}})$ with a smaller or equal KL divergence. Let's choose $\gamma_{\text{DS}} = \frac{\beta_{\text{DS}}}{\beta_{\text{ent}}}\gamma_{\text{ent}}$, and set $C(\pi_{\text{DS}}) = C(\pi_{\text{ent}}) = C$. Equivalently, we assume that

$$\frac{\gamma_{\text{DS}}}{\beta_{\text{DS}}} = \frac{\gamma_{\text{ent}}}{\beta_{\text{ent}}} = \widetilde{\gamma}.$$

Then the KL divergence for our method becomes: According to Jensen's inequality, we have

$$\frac{\widetilde{\gamma} \sum_{\tau \notin \mathcal{C}} \pi_{\text{base}}(\tau)^{1-\widetilde{\gamma}} \ln(\pi_{\text{base}}(\tau))}{B_{\widetilde{\gamma}} - b_{\widetilde{\gamma}}} \cdot (1 - C) \le (1 - C) \cdot \ln\left(\frac{\sum_{\tau \notin \mathcal{C}} \pi_{\text{base}}(\tau)}{B_{\widetilde{\gamma}} - b_{\widetilde{\gamma}}}\right) = (1 - C) \cdot \ln\left(\frac{1 - p_C}{B_{\widetilde{\gamma}} - b_{\widetilde{\gamma}}}\right)$$

Then in this case we have

$$\mathbb{D}_{\text{KL}}(\pi_{\text{ent}}\|\pi_{\text{base}}) = -\frac{\widetilde{\gamma} \cdot \sum_{\tau \in \mathcal{C}} \pi_{\text{base}}(\tau)^{1-\widetilde{\gamma}} \ln(\pi_{\text{base}}(\tau))}{b_{\widetilde{\gamma}}} \cdot C + (1 - C)\left[\ln\left(\frac{b_{\widetilde{\gamma}}}{B_{\widetilde{\gamma}} - b_{\widetilde{\gamma}}}\right)\right] - \ln b_{\widetilde{\gamma}}$$

$$+ C \ln C + (1 - C) \ln(1 - C) - \frac{\widetilde{\gamma} \sum_{\tau \notin \mathcal{C}} \pi_{\text{base}}(\tau)^{1-\widetilde{\gamma}} \ln(\pi_{\text{base}}(\tau))}{1 - p_C} \cdot (1 - C)$$

$$\ge -\frac{\widetilde{\gamma} \sum_{\tau \in \mathcal{C}} \pi_{\text{base}}(\tau)^{1-\widetilde{\gamma}} \ln(\pi_{\text{base}}(\tau))}{b_{\widetilde{\gamma}}} \cdot C + (1 - C)\left[\ln\left(\frac{b_{\widetilde{\gamma}}}{B_{\widetilde{\gamma}} - b_{\widetilde{\gamma}}}\right)\right] - \ln b_{\widetilde{\gamma}}$$

$$+ C \ln C + (1 - C) \ln(1 - C)$$

$$= \mathbb{D}_{\text{KL}}(\pi_{\text{DS}}\|\pi_{\text{base}})$$

Therefore, the difference is non-negative: $\mathbb{D}_{\text{KL}}(\pi_{\text{ent}}\|\pi_{\text{base}}) - \mathbb{D}_{\text{KL}}(\pi_{\text{DS}}\|\pi_{\text{base}}) \ge 0$. This means that for any given correctness level $C$, our method (with $\gamma_{\text{DS}} = \gamma_{\text{ent}} \cdot \frac{\beta_{\text{DS}}}{\beta_{\text{ent}}}$) can achieve it with a lower or equal KL-divergence cost. Thus, if the entropy method is constrained by $\mathbb{D}_{\text{KL}} \le \kappa$, our method can achieve a correctness $C(\pi_{\text{DS}}) \ge C(\pi_{\text{ent}})$ while also satisfying the constraint. $\qquad\square$

**Lemma 6** (Correctness under Reverse $\chi^2$ Constraint). *When $K_\rho(\pi, \pi_{base}) = \mathbb{D}_{\chi^2}(\pi_{base}\|\pi)$, for any $\gamma_{ent} \ge 0, \beta_{ent} > 0$ such that $\mathbb{D}_{KL}(\pi_{base}\|\pi_{ent}) \le \kappa$, there exist $\gamma_{DS} \ge 0$ and $\beta_{DS} > 0$ such that $\mathbb{D}_{\chi^2}(\pi_{base}\|\pi_{DS}) \le \kappa$ and $C(\pi_{DS}) \ge C(\pi_{ent})$.*

**Step 1: Analyze the Entropy-Maximization Policy ($\pi_{\mathbf{ent}}$).** The correctness of $\pi_{\text{ent}}$ is the total probability mass on correct trajectories:

$$C(\pi_{\text{ent}}) = \frac{\sum_{\tau \in \mathcal{C}} [\pi_{\text{base}}(\tau)]^{1 - \frac{\gamma_{\text{ent}}}{\beta_{\text{ent}}}} \exp(1/\beta_{\text{ent}})}{\sum_{\tau \in \mathcal{T}} [\pi_{\text{base}}(\tau)]^{1 - \frac{\gamma_{\text{ent}}}{\beta_{\text{ent}}}} \exp(r(\tau)/\beta_{\text{ent}})} = \frac{b_{\frac{\gamma_{\text{ent}}}{\beta_{\text{ent}}}} e^{1/\beta_{\text{ent}}}}{b_{\frac{\gamma_{\text{ent}}}{\beta_{\text{ent}}}} e^{1/\beta_{\text{ent}}} + (B_{\frac{\gamma_{\text{ent}}}{\beta_{\text{ent}}}} - b_{\frac{\gamma_{\text{ent}}}{\beta_{\text{ent}}}})}. \tag{10}$$

Solving for $e^{1/\beta_{\text{ent}}}$ yields: $e^{1/\beta_{\text{ent}}} = \frac{B_{\frac{\gamma_{\text{ent}}}{\beta_{\text{ent}}}} - b_{\frac{\gamma_{\text{ent}}}{\beta_{\text{ent}}}}}{b_{\frac{\gamma_{\text{ent}}}{\beta_{\text{ent}}}}} \left(\frac{C(\pi_{\text{ent}})}{1 - C(\pi_{\text{ent}})}\right)$.

The reverse $\chi^2$ divergence $\mathbb{D}_{\chi^2}(\pi_{\text{base}}\|\pi_{\text{ent}})$ can be expressed as a function of $C(\pi_{\text{ent}})$. Following the derivation previously, we arrive at:

$$\mathbb{D}_{\chi^2}(\pi_{\text{base}}\|\pi_{\text{ent}}) = \sum_\tau \pi_{\text{base}}(\tau)\left(\frac{\pi_{\text{ent}}(\tau)}{\pi_{\text{base}}(\tau)} - 1\right)^2 = \sum_\tau \frac{(\pi_{\text{ent}}(\tau))^2}{\pi_{\text{base}}(\tau)} - 1$$

We insert the expression of $\pi_{\text{ent}}$ into the divergence constraints and we can obtain that

$$\mathbb{D}_{\chi^2}(\pi_{\text{base}}\|\pi_{\text{ent}}) = -1 + \frac{e^{\frac{2}{\beta_{\text{ent}}}}\sum_{\tau\in\mathcal{C}}\pi_{\text{base}}(\tau)^{1-2\frac{\gamma_{\text{ent}}}{\beta_{\text{ent}}}} + \sum_{\tau\notin\mathcal{C}}\pi_{\text{base}}(\tau)^{1-2\frac{\gamma_{\text{ent}}}{\beta_{\text{ent}}}}}{\left[B_{\frac{\gamma_{\text{ent}}}{\beta_{\text{ent}}}} - b_{\frac{\gamma_{\text{ent}}}{\beta_{\text{ent}}}} + b_{\frac{\gamma_{\text{ent}}}{\beta_{\text{ent}}}}e^{\frac{1}{\beta_{\text{ent}}}}\right]^2}$$

We then insert Eq. 10 into the expression of divergence and we can obtain that

$$\mathbb{D}_{\chi^2}(\pi_{\text{base}}\|\pi_{\text{ent}}) = -1 + \frac{\sum_{\tau\in\mathcal{C}}\pi_{\text{base}}(\tau)^{1-2\frac{\gamma_{\text{ent}}}{\beta_{\text{ent}}}}}{b_{\frac{\gamma_{\text{ent}}}{\beta_{\text{ent}}}}^2}\cdot C(\pi_{\text{ent}})^2 + (1-C(\pi_{\text{ent}}))^2\frac{\sum_{\tau\notin\mathcal{C}}\pi_{\text{base}}(\tau)^{1-2\frac{\gamma_{\text{ent}}}{\beta_{\text{ent}}}}}{\left(B_{\frac{\gamma_{\text{ent}}}{\beta_{\text{ent}}}} - b_{\frac{\gamma_{\text{ent}}}{\beta_{\text{ent}}}}\right)^2}$$

**Step 2: Analyze Differential Policy ($\pi_{\text{DS}}$).** Similarly, the correctness is:

$$C(\pi_{\text{DS}}) = \frac{b_{\frac{\gamma_{\text{DS}}}{\beta_{\text{DS}}}}e^{1/\beta_{\text{DS}}}}{b_{\frac{\gamma_{\text{DS}}}{\beta_{\text{DS}}}}e^{1/\beta_{\text{DS}}} + (1-p_c)}.$$

The reverse $\chi^2$ divergence, after a similar derivation, is:

$$\mathbb{D}_{\chi^2}(\pi_{\text{base}}\|\pi_{\text{DS}}) = -1 + \frac{\sum_{\tau\in\mathcal{C}}\pi_{\text{base}}(\tau)^{1-2\frac{\gamma_{\text{DS}}}{\beta_{\text{DS}}}}}{b_{\frac{\gamma_{\text{DS}}}{\beta_{\text{DS}}}}^2}\cdot C(\pi_{\text{DS}})^2 + (1-C(\pi_{\text{DS}}))^2\frac{\sum_{\tau\notin\mathcal{C}}\pi_{\text{base}}(\tau)}{(1-p_c)^2}$$

**Step 3: Compare the Policies.** Our goal is to show that for any $C(\pi_{\text{ent}})$, we can choose parameters for our method to achieve $C(\pi_{\text{DS}}) \geq C(\pi_{\text{ent}})$ with a smaller or equal $\chi^2$ divergence. Let's choose $\gamma_{\text{DS}} = \frac{\beta_{\text{DS}}}{\beta_{\text{ent}}}\gamma_{\text{ent}}$, and set $C(\pi_{\text{DS}}) = C(\pi_{\text{ent}}) = C$. Equivalently, we assume that

$$\frac{\gamma_{\text{DS}}}{\beta_{\text{DS}}} = \frac{\gamma_{\text{ent}}}{\beta_{\text{ent}}} = \widetilde{\gamma}.$$

According to Cauchy Inequality:

$$\left[\sum_{\tau\notin\mathcal{C}}\pi_{\text{base}}(\tau)^{1-2\widetilde{\gamma}}\right]\cdot\left[\sum_{\tau\notin\mathcal{C}}\pi_{\text{base}}(\tau)\right] \geq \left[\sum_{\tau\notin\mathcal{C}}\pi_{\text{base}}(\tau)^{1-\widetilde{\gamma}}\right]^2.$$

Thus, we have

$$\mathbb{D}_{\chi^2}(\pi_{\text{base}}\|\pi_{\text{ent}}) = -1 + \frac{\sum_{\tau\in\mathcal{C}}\pi_{\text{base}}(\tau)^{1-2\widetilde{\gamma}}}{b_{\widetilde{\gamma}}^2}\cdot C(\pi_{\text{ent}})^2 + (1-C(\pi_{\text{ent}}))^2\frac{\sum_{\tau\notin\mathcal{C}}\pi_{\text{base}}(\tau)^{1-2\widetilde{\gamma}}}{(B_{\widetilde{\gamma}} - b_{\widetilde{\gamma}})^2}$$

$$\geq -1 + \frac{\sum_{\tau\in\mathcal{C}}\pi_{\text{base}}(\tau)^{1-2\widetilde{\gamma}}}{b_{\widetilde{\gamma}}^2}\cdot C(\pi_{\text{ent}})^2 + (1-C(\pi_{\text{ent}}))^2\frac{1}{\sum_{\tau\notin\mathcal{C}}\pi_{\text{base}}(\tau)} = \mathbb{D}_{\chi^2}(\pi_{\text{base}}\|\pi_{\text{DS}}).$$

Therefore, the difference is non-negative: $\mathbb{D}_{\chi^2}(\pi_{\text{base}}\|\pi_{\text{ent}}) - \mathbb{D}_{\chi^2}(\pi_{\text{base}}\|\pi_{\text{DS}}) \geq 0$. This means that for any given correctness level $C$, our method (with $\gamma_{\text{DS}} = \gamma_{\text{ent}}\cdot\frac{\beta_{\text{DS}}}{\beta_{\text{ent}}}$) can achieve it with a lower or equal KL-divergence cost. Thus, if both methods is constrained by $\mathbb{D}_{\text{KL}}(\pi_{\text{base}}\|\pi_{\text{ent}}) \leq \kappa$ and $\mathbb{D}_{\text{KL}}(\pi_{\text{base}}\|\pi_{\text{DS}}) \leq \kappa$, our method can achieve a correctness $C(\pi_{\text{DS}}) \geq C(\pi_{\text{ent}})$.

**Lemma 7** (Correctness under Forward $\chi^2$ Constraint). *When $K_\rho(\pi, \pi_{base}) = \mathbb{D}_{\chi^2}(\pi\|\pi_{base})$, for any $\gamma_{ent} \geq 0, \beta_{ent} > 0$ such that $\mathbb{D}_{\chi^2}(\pi_{ent}\|\pi_{base}) \leq \kappa$, there exist $\gamma_{DS} \geq 0$ and $\beta_{DS} > 0$ such that $\mathbb{D}_{\chi^2}(\pi_{DS}\|\pi_{base}) \leq \kappa$ and $C(\pi_{DS}) \geq C(\pi_{ent})$.*

*Proof.* The proof proceeds in three steps: we first find a functional relationship between correctness $C$ and the KL divergence for each policy, and then compare them.

**Step 1: Analyze the Entropy-Maximization Policy ($\pi_{\text{ent}}$).** The correctness of $\pi_{ent}$ is the total probability mass on correct trajectories:

$$C(\pi_{\text{ent}}) = \frac{\sum_{\tau\in\mathcal{C}}\left[\pi_{\text{base}}(\tau)\right]^{1-\frac{\gamma_{\text{ent}}}{\beta_{\text{ent}}}}\exp(1/\beta_{\text{ent}})}{\sum_{\tau\in\mathcal{T}}\left[\pi_{\text{base}}(\tau)\right]^{1-\frac{\gamma_{\text{ent}}}{\beta_{\text{ent}}}}\exp(r(\tau)/\beta_{\text{ent}})} = \frac{b_{\frac{\gamma_{\text{ent}}}{\beta_{\text{ent}}}}e^{1/\beta_{\text{ent}}}}{b_{\frac{\gamma_{\text{ent}}}{\beta_{\text{ent}}}}e^{1/\beta_{\text{ent}}} + (B_{\frac{\gamma_{\text{ent}}}{\beta_{\text{ent}}}} - b_{\frac{\gamma_{\text{ent}}}{\beta_{\text{ent}}}})}. \quad (11)$$

Solving for $e^{1/\beta_{\text{ent}}}$ yields: $e^{1/\beta_{\text{ent}}} = \frac{B_{\frac{\gamma_{\text{ent}}}{\beta_{\text{ent}}}} - b_{\frac{\gamma_{\text{ent}}}{\beta_{\text{ent}}}}}{b_{\frac{\gamma_{\text{ent}}}{\beta_{\text{ent}}}}} \left( \frac{C(\pi_{\text{ent}})}{1 - C(\pi_{\text{ent}})} \right).$

The reverse $\chi^2$ divergence $\mathbb{D}_{\chi^2}(\pi_{\text{ent}} \| \pi_{\text{base}})$ can be expressed as a function of $C(\pi_{\text{ent}})$. Following the derivation previously, we arrive at:

$$\mathbb{D}_{\chi^2}(\pi_{\text{ent}} \| \pi_{\text{base}}) = -1 + \left[ \sum_\tau \pi_{\text{base}}(\tau)^{1 - \frac{\gamma_{\text{ent}}}{\beta_{\text{ent}}}} \exp\left( \frac{r(\tau)}{\beta_{\text{ent}}} \right) \right] \cdot \left[ \sum_\tau \pi_{\text{base}}(\tau)^{1 + \frac{\gamma_{\text{ent}}}{\beta_{\text{ent}}}} \exp\left( -\frac{r(\tau)}{\beta_{\text{ent}}} \right) \right]$$

$$= \frac{1}{C(\pi_{\text{ent}})} b_{\frac{\gamma_{\text{ent}}}{\beta_{\text{ent}}}} \sum_{\tau \in \mathcal{C}} \pi_{\text{base}}(\tau)^{1 + \frac{\gamma_{\text{ent}}}{\beta_{\text{ent}}}} + \frac{1}{1 - C(\pi_{\text{ent}})} (B_{\frac{\gamma_{\text{ent}}}{\beta_{\text{ent}}}} - b_{\frac{\gamma_{\text{ent}}}{\beta_{\text{ent}}}}) \sum_{\tau \notin \mathcal{C}} \pi_{\text{base}}(\tau)^{1 - \frac{\gamma_{\text{ent}}}{\beta_{\text{ent}}}}$$

**Step 2: Analyze Differential Policy ($\pi_{\text{DS}}$).** Similarly, the correctness for our policy is given by:

$$C(\pi_{\text{DS}}) = \frac{b_{\frac{\gamma_{\text{DS}}}{\beta_{\text{DS}}}} e^{1/\beta_{\text{DS}}}}{b_{\frac{\gamma_{\text{DS}}}{\beta_{\text{DS}}}} e^{1/\beta_{\text{DS}}} + (1 - p_c)}. \tag{12}$$

The corresponding reverse $\chi^2$ divergence as a function of $C(\pi_{\text{DS}})$ is:

$$\mathbb{D}_{\chi^2}(\pi_{\text{ent}} \| \pi_{\text{base}}) = \frac{1}{C(\pi_{\text{ent}})} b_{\frac{\gamma_{\text{DS}}}{\beta_{\text{DS}}}} \sum_{\tau \in \mathcal{C}} \pi_{\text{base}}(\tau)^{1 + \frac{\gamma_{\text{DS}}}{\beta_{\text{DS}}}} + \frac{1}{(1 - C(\pi_{\text{ent}}))} ((1 - p_c)) \sum_{\tau \notin \mathcal{C}} \pi_{\text{base}}(\tau)^{1 + \frac{\gamma_n}{\beta_{\text{DS}}}}$$

**Step 3: Compare the Policies.** Our goal is to show that for any $C(\pi_{\text{ent}})$, we can choose parameters for our method to achieve $C(\pi_{\text{DS}}) \geq C(\pi_{\text{ent}})$ with a smaller or equal $\chi^2$ divergence. Let's choose $\gamma_{\text{DS}} = \frac{\beta_{\text{DS}}}{\beta_{\text{ent}}} \gamma_{\text{ent}}$, and set $C(\pi_{\text{DS}}) = C(\pi_{\text{ent}}) = C$. Equivalently, we assume that

$$\frac{\gamma_{\text{DS}}}{\beta_{\text{DS}}} = \frac{\gamma_{\text{ent}}}{\beta_{\text{ent}}} = \widetilde{\gamma}.$$

. According to Cauchy Inequality:

$$\left[ \sum_{\tau \notin \mathcal{C}} \pi_{\text{base}}(\tau)^{1 + \widetilde{\gamma}} \right] \cdot \left[ \sum_{\tau \notin \mathcal{C}} \pi_{\text{base}}(\tau)^{1 - \widetilde{\gamma}} \right] \geq \left[ \sum_{\tau \notin \mathcal{C}} \pi_{\text{base}}(\tau) \right]^2.$$

Thus, we have

$$\mathbb{D}_{\chi^2}(\pi_{\text{ent}} \| \pi_{\text{base}}) = \frac{1}{C(\pi_{\text{ent}})} b_{\widetilde{\gamma}} \sum_{\tau \in \mathcal{C}} \pi_{\text{base}}(\tau)^{1 + \widetilde{\gamma}} + \frac{1}{1 - C(\pi_{\text{ent}})} (B_{\widetilde{\gamma}} - b_{\widetilde{\gamma}}) \sum_{\tau \notin \mathcal{C}} \pi_{\text{base}}(\tau)^{1 - \widetilde{\gamma}}$$

$$\geq \frac{1}{C(\pi_{\text{ent}})} b_{\widetilde{\gamma}} \sum_{\tau \in \mathcal{C}} \pi_{\text{base}}(\tau)^{1 + \widetilde{\gamma}} + \frac{1}{1 - C(\pi_{\text{ent}})} \left( \sum_{\tau \notin \mathcal{C}} \pi_{\text{base}}(\tau) \right)^2 = \mathbb{D}_{\chi^2}(\pi_{\text{ent}} \| \pi_{\text{base}})$$

Therefore, the difference is non-negative: $\mathbb{D}_{\chi^2}(\pi_{\text{ent}} \| \pi_{\text{base}}) - \mathbb{D}_{\chi^2}(\pi_{\text{DS}} \| \pi_{\text{base}}) \geq 0$. This means that for any given correctness level $C$, our method (with $\gamma_{\text{DS}} = \gamma_{\text{ent}} \cdot \frac{\beta_{\text{DS}}}{\beta_{\text{ent}}}$) can achieve it with a lower or equal KL-divergence cost. Thus, if the entropy method is constrained by $\mathbb{D}_{\chi^2} \leq \kappa$, our method can achieve a correctness $C(\pi_{\text{DS}}) \geq C(\pi_{\text{ent}})$ while also satisfying the constraint. $\qquad \square$

**Theorem B.1.** *Assume the reward mechanism has access to all correct trajectories. For any parameters $\gamma_{ent} \geq 0$ and $\beta_{ent} > 0$ used in the entropy-regularized policy $\pi_{ent}$ that satisfy a proximity constraint $K_\rho(\pi_{ent}, \pi_{base}) \leq \kappa$, there exist parameters $\gamma_{DS} \geq 0$ and $\beta_{n,p} > 0$ for our proposed policy $\pi_{DS}$ such that it also satisfies $K_\rho(\pi_{DS}, \pi_{base}) \leq \kappa$, and the following inequalities hold:*

$$C(\pi_{DS}) \geq C(\pi_{ent}) \quad and \quad \sigma_{DS} \geq \sigma_{En}.$$

*This result holds for divergence measures $K_\rho(\pi, \pi_{base})$ including $\mathbb{D}_{KL}(\pi \| \pi_{base})$, $\mathbb{D}_{KL}(\pi_{base} \| \pi)$, $\mathbb{D}_{\chi^2}(\pi \| \pi_{base})$, and $\mathbb{D}_{\chi^2}(\pi_{base} \| \pi)$.*

*Proof.* According to Lemma 4, Lemma 5, Lemma 6, and Lemma 7, we obtain that Theorem B.1 holds for $K_\rho(\pi, \pi_{base}) = \mathbb{D}_{KL}(\pi \| \pi_{base}), \mathbb{D}_{KL}(\pi_{base} \| \pi), \mathbb{D}_{\chi^2}(\pi \| \pi_{base}), \mathbb{D}_{\chi^2}(\pi_{base} \| \pi)$. Thus, we finish the proof of the theorem. $\qquad \square$

### B.4 EQUIVALENCE OF THEORETICAL AND PRACTICAL REWARD MODIFICATIONS

In this section, we clarify the relationship between our theoretical reward modification and its practical implementation. Specifically, we demonstrate that subtracting a $\log \pi$ term from the reward is equivalent to subtracting a $\log \pi_{\text{base}}$ term, under a re-parameterization of the optimization objective.

Consider the following theoretical reward modification, which uses the policy's own probability $\pi$:

$$r_{\text{DS}}^{\pi}(\tau) = \begin{cases} r(\tau) - \gamma_p \cdot \log(\pi(\tau)) & \text{if } r(\tau) > 0 \quad \text{(correct trajectories)} \\ r(\tau) + \gamma_n \cdot \log(\pi(\tau)) & \text{if } r(\tau) \leq 0 \quad \text{(incorrect trajectories).} \end{cases} \tag{13}$$

We first define the theoretical optimization problem for parameters $\beta, \gamma_n, \gamma_p$:

$$\pi_{\text{DS}} = \arg\max_{\pi} \mathbb{E}_{\tau \sim \pi} \left[ r_{\text{DS}}^{\pi}(\tau) \right] - \beta \cdot \mathbb{D}_{\text{KL}}(\pi || \pi_{\text{base}}). \tag{14}$$

Now, consider an alternative formulation where the reward is modified using the base policy $\pi_{\text{base}}$:

$$r_{\text{DS}}^{\pi_{\text{base}}}(\tau) = \begin{cases} r(\tau) - \widetilde{\gamma}_p \cdot \log(\pi_{\text{base}}(\tau)) & \text{if } r(\tau) > 0 \quad \text{(correct trajectories)} \\ r(\tau) + \widetilde{\gamma}_n \cdot \log(\pi_{\text{base}}(\tau)) & \text{if } r(\tau) \leq 0 \quad \text{(incorrect trajectories).} \end{cases} \tag{15}$$

We show that the solution $\pi_{\text{DS}}$ to the original problem equation 14 is **also** the solution to the following practical objective, which uses $r_{\text{DS}}^{\pi_{\text{base}}}$:

$$\pi_{\text{DS}} = \arg\max_{\pi} \mathbb{E}_{\tau \sim \pi} \left[ r_{\text{DS}}^{\pi_{\text{base}}}(\tau) \right] - \widetilde{\beta} \cdot \mathbb{D}_{\text{KL}}(\pi || \pi_{\text{base}}). \tag{16}$$

This equivalence holds when the new parameters $\widetilde{\beta}, \widetilde{\gamma}_p$, and $\widetilde{\gamma}_n$ are set as follows:

$$\widetilde{\beta} = \beta + \gamma_p, \quad \widetilde{\gamma}_p = \gamma_p, \quad \widetilde{\gamma}_n = \frac{\gamma_n(\beta + \gamma_p)}{\beta + \gamma_n}. \tag{17}$$

Therefore, the theoretical analysis from Theorem 6.1 still holds when $\log \pi$ is substituted for $\log \pi_{\text{base}}$. Furthermore, the policy selection mechanism in Eq. 2 is equivalent to directly maximizing entropy via an added regularization term. In practice, we find that using $\log \pi_{\theta_{\text{old}}}$ (i.e., the log-probability of a previous policy iteration) in the advantage function modification yields empirically better performance than using $\log \pi_{\text{base}}$ (the log-probability of the base policy).

## C EXPERIMENTAL DETAILS

In this section, we provide additional details for the experiments in Section 4.

### C.1 COUNTDOWN EXPERIMENT

#### C.1.1 DATA

We use the dataset released by Pan et al. (2025), which contains 327,680 training samples and 1,024 test samples.[1] An example training prompt is shown below.

---

**Countdown Task Example**

**[INST]** Using the numbers [5, 94, 9, 44], create an equation that equals 93. You can use basic arithmetic operations (+, -, *, /) and each number can only be used once. Show your work in `<think> </think>` tags. And return the final answer in `<answer> </answer>` tags, for example `<answer>(1 + 2) / 3</answer>`. **[/INST]** Let me solve this step by step.

---

Our implementation builds on the official repository of Pan et al. (2025)[2] and a fork adapted for A100 training.[3]

---

[1] https://huggingface.co/datasets/Jiayi-Pan/Countdown-Tasks-3to4
[2] https://github.com/Jiayi-Pan/TinyZero
[3] https://github.com/JerryWu-code/TinyZero

### C.1.2 TRAINING

We train with a global batch size of $128$, with $5$ rollouts per prompt, and use a mini-batch size of $64$. The learning rate is $1 \times 10^{-6}$, and the KL penalty coefficient is set to $\beta_{\mathrm{KL}} = 1 \times 10^{-3}$. The reward is $1$ for correct responses, $0.1$ for incorrect yet properly formatted responses, and $0$ for all others. The maximum response length is $8{,}192$ tokens. We perform RL fine-tuning of the `Qwen2.5-3B-Instruct` model (Qwen Team, 2024) for 320 steps on 2 A100 GPUs.

*Table A1: Configuration for Qwen3-1.7B*

| Parameter | Value | Parameter | Value |
|---|---|---|---|
| Pretrained model | Qwen3-1.7B | Training set | DAPO14k |
| Prompts per batch | 32 | Generations per prompt | 8 |
| Gradient update per RL step | 2 | Max prompt length | 1024 |
| Max response length | 4096 | Learning rate | $5 \times 10^{-7}$ |
| Clip ratio low | 0.2 | Clip ratio high | 0.25 |
| Training Steps | 300 | $\beta$ | 0.0 |
| Entropy coefficient | 0.0 | $\gamma_p$ | 0.02 |
| $\gamma_n$ | 0.002 | Remove padding | Enabled |
| Rollout engine | vllm | Rollout temperature | 0.7 |
| Validation temperature | 0.7 | Device | 4 x Nvidia-H100 |

*Table A2: Configuration for Qwen2.5-Math-1.5B*

| Parameter | Value | Parameter | Value |
|---|---|---|---|
| Pretrained Model | Qwen2.5-Math-1.5B | Training Set | DAPO14k + MATH12k |
| Prompts per batch | 32 | Generations per prompt | 8 |
| Gradient update per RL step | 1 | Max prompt length | 1024 |
| Max response length | 2048 | Learning rate | $1 \times 10^{-6}$ |
| Clip ratio low | 0.2 | Clip ratio high | 0.25 |
| Training Steps | 1000 | $\beta$ | 0.0 |
| Entropy coefficient | 0.0 | $\gamma_p$ | 0.01 |
| $\gamma_n$ | 0.01 | Remove padding | Enabled |
| Rollout engine | vllm | Rollout temperature | 0.7 |
| Validation temperature | 0.7 | Device | 4 x Nvidia-L6000 |

*Table A3: Configuration for Qwen2.5-Math-7B*

| Parameter | Value | Parameter | Value |
|---|---|---|---|
| Pretrained Model | Qwen2.5-Math-7B | Training Set | DAPO14k + MATH12k |
| Prompts per batch | 32 | Generations per prompt | 8 |
| Gradient update per RL step | 1 | Max prompt length | 1024 |
| Max response length | 2048 | Learning rate | $1 \times 10^{-6}$ |
| Clip ratio low | 0.2 | Clip ratio high | 0.25 |
| Training Steps | 500 | $\beta$ | 0.0 |
| Entropy coefficient | 0.0 | $\gamma_p$ | 0.01 |
| $\gamma_n$ | 0.01 | Remove padding | Enabled |
| Rollout engine | vllm | Rollout temperature | 0.7 |
| Validation temperature | 0.7 | Device | 4 x Nvidia-A100 |

For Qwen2.5-Math-7B model, we trained for three random seeds. During evaluation, we first generated 128 rollouts for each question, then estimated Pass@1 to Pass@64 using the unbiased estimator of each metric respectively, following Walder & Karkhanis (2025b).

*Table A4: Configuration for Ministral-8B-Instruct*

| Parameter | Value | Parameter | Value |
|---|---|---|---|
| Pretrained model | Ministral-8B-Instruct | Training set | DAPO14k + MATH12k |
| Prompts per batch | 32 | Generations per prompt | 8 |
| Gradient update per RL step | 2 | Max prompt length | 1024 |
| Max response length | 2048 | Learning rate | $3 \times 10^{-7}$ |
| Clip ratio low | 0.2 | Clip ratio high | 0.22 |
| Training steps | 300 | $\beta$ | 0.001 |
| Entropy coefficient | 0.0 | $\gamma_p$ | 0.02 |
| $\gamma_n$ | 0.002 | Remove padding | Enabled |
| Rollout engine | `vllm` | Rollout temperature | 0.7 |
| Validation temperature | 0.7 | Device | 4 x Nvidia-A100 |

## C.2 BASELINE IMPLEMENTATION

**GR-PKPO.** We attempted to train the model using the Pass@$k$ metric directly as the reward signal for $k \in \{2, 3, 4\}$. However, this approach proved unstable across all configurations. The training process quickly collapsed, causing the model to generate degenerate outputs and yielding performance substantially worse than the baseline. Consequently, these results are omitted from our main comparisons. We hypothesize this instability may be attributed to the limited number of rollouts (5) used during training.

**Unlikeliness Reward Method.** We compare against the rank-based penalty proposed by He et al. (2025a), which down-weights the reward for high-probability solutions to encourage diversity. For a set of rollouts $\{y_i\}_{i=1}^{G}$, the modified reward is:

$$r_{\text{unlikely}}(y_i) = r(y_i) \left( 1 - \beta_{\text{rank}} \frac{G - \text{rank}(y_i)}{G} \right),$$

where $\text{rank}(y_i)$ is the rank of the solution $y_i$ based on its probability under the policy $\pi_{\text{old}}(\cdot \mid x)$. We performed a hyperparameter sweep for $\beta_{\text{rank}} \in \{0.05, 0.1, 0.15, 0.2, 0.25, 0.3\}$ and found that $\beta_{\text{rank}} = 0.05$ yielded the best Pass@64 performance. For values of $\beta_{\text{rank}} \geq 0.15$, the training became unstable and collapsed.

**Entropy-Based Methods.** For the entropy-based baselines, we augment GRPO with either an entropy bonus or penalty. We swept the coefficients $\eta_+$ and $\eta_-$ over the set $\{0.005, 0.01, 0.03, 0.05\}$. In Figure 4, we report the results for the hyperparameter value that achieved the highest Pass@64 score. A comprehensive summary of all experimental runs is available in Appendix D.

## C.3 MATH REASONING EXPERIMENT

We first filter out 14,042 questions that are written in English from the DAPO dataset (Yu et al., 2025), and combined them with 11,998 questions in the MATH12k dataset (Hendrycks et al., 2021). After that, we conduct reinforcement finetuning on `Qwen2.5-Math-1.5B` (Qwen Team, 2024), `Qwen3-1.7B` (Qwen Team, 2025) and `Ministral-8B-Instruct` (Jiang et al., 2024) model. The detailed training setups are shown in Tab. A6A4A5, and response examples are shown in boxes below respectively.

---

**Mistral Instruction Response Example**

**[INST]** Convert the point $(0, 3)$ in rectangular coordinates to polar coordinates. Enter your answer in the form $(r, \theta)$, where $r > 0$ and $0 \leq \theta < 2\pi$. Let's think step by step and output the final answer within `\boxed{}`. **[/INST]**
1. **Calculate the radius $r$:**

$$r = \sqrt{0^2 + 3^2} = \sqrt{0 + 9} = \sqrt{9} = 3.$$

2. **Determine the angle $\theta$:**
   - The point $(0, 3)$ is on the positive $y$-axis.

---

- The angle $\theta$ with the positive $x$-axis is given by:

$$\theta = \tan^{-1}\left(\frac{3}{0}\right).$$

- Since the point is on the positive $y$-axis, the angle is $\frac{\pi}{2}$.

3. **Combine the results to get the polar coordinates:**

$$\boxed{\left(3, \frac{\pi}{2}\right)}$$

---

**Qwen2.5-Math-1.5B**

**[INST]** Cities $A$ and $B$ are $45$ miles apart. Alicia lives in $A$ and Beth lives in $B$. Alicia bikes towards $B$ at 18 miles per hour. Leaving at the same time, Beth bikes toward $A$ at 12 miles per hour. How many miles from City $A$ will they be when they meet? Let's think step by step and output the final answer within \boxed{}. **[/INST]**
To determine where Alicia and Beth will meet, we need to find out how long it will take for them to meet and then use that time to calculate the distance Alicia has traveled.

1. First, let's calculate their combined speed. Since Alicia bikes at 18 miles per hour and Beth bikes at 12 miles per hour, their combined speed is $18 + 12 = 30$ miles per hour.

2. Since the distance between the two cities is 45 miles, the time it takes for them to meet is the total distance divided by their combined speed. So, the time $t$ it takes for them to meet is

$$t = \frac{45}{30} = 1.5 \text{ hours.}$$

3. Now, we need to find out how far Alicia has traveled in 1.5 hours. Since Alicia bikes at 18 miles per hour, the distance she travels in 1.5 hours is

$$18 \times 1.5 = 27 \text{ miles.}$$

So, Alicia and Beth will meet 27 miles from City $A$. The final answer is

$$\boxed{27}$$

---

*Table A5: Configuration for Countdown Task*

| Parameter | Value | Parameter | Value |
|-----------|-------|-----------|-------|
| Pretrained model | Qwen2.5-3B-Instruct | Batch size | 128 |
| Generations per prompt | 5 | Mini-batch size | 64 |
| Max prompt length | 2,048 | Max response length | 8,192 |
| Learning rate | $1 \times 10^{-6}$ | Training steps | 320 |
| Entropy coefficient | 0.001 | Clip ratio | 0.2 |
| $\gamma_p$ | 0.03 | $\gamma_n$ | 0.01 |
| Rollout engine | vllm | Rollout temperature | 1 |
| Validation temperature | 1 | Validation top-k | 50 |
| Validation top-p | 0.7 | Device | 2 x A100 |

## C.4 EXPERIMENTAL DETAILS FOR SECTION 5.2

In this section, we present experimental details to substantiate the claim made in Section 5.2: namely, that the Solution Multiplicity Solution Multiplicity$(\mathcal{X})$, the average number of unique solutions per

problem instance, serves as an appropriate metric for evaluating the benefits of encouraging policy entropy in a task. The tasks we consider are as follows:

- **Math Reasoning**: As described in Appendix C.3.

- **Countdown**: As described in Appendix C.1.

- **Countdown-3**: A variant of Countdown where we filter the dataset to retain only samples with three given numbers.

- **Knights and Knaves**: A classic logic puzzle in which the objective is to identify truth-tellers (knights) and liars (knaves) from their statements. (Stojanovski et al., 2025). Below is an example question and correct response.

---

### Knights-and-Knaves

**Question:**
A very special island is inhabited only by sages and fools. Sages always tell the truth, and fools always lie. You meet 6 inhabitants: Owen, Liam, Evelyn, William, Sofia, and Emily. As Owen put it, "William is a sage". Liam was heard saying, "William is a fool". Evelyn noted, "Sofia is a fool and Evelyn is a sage". "if Owen is a sage then Sofia is a sage," William mentioned. "if Liam is a sage then Evelyn is a sage," Sofia claimed. Emily told you that if Emily is a sage then Liam is a fool. So who is a sage and who is a fool? (Format your answer like: "Owen is a sage/fool, Liam is a sage/fool, Evelyn is a sage/fool, William is a sage/fool, Sofia is a sage/fool, and Emily is a sage/fool") Let's think step by step and output the final answer (should be a sentence) within \boxed{}.

**Response:**

- We need to consider two cases for Owen:
    - *Case 1: Owen is a sage.* If Owen is a sage, then his statement is true. Therefore, William must be a sage.
    - *Case 2: Owen is a fool.* If Owen is a fool, then his statement is false. Therefore, William must be a fool.
- **Liam's statement**: "William is a fool."
    - *Case 1: Liam is a sage.* If Liam is a sage, then his statement is true. Therefore, William must be a fool.
    - *Case 2: Liam is a fool.* If Liam is a fool, then his statement is false. Therefore, William must be a sage.
- **William's statement**: "If Owen is a sage then Sofia is a sage."
  Since Sofia is a sage, William's statement is true. Therefore, William must be a sage.
- **Owen's statement revisited**: "William is a sage."
  Since William is a sage, Owen's statement is true. Therefore, Owen must be a sage.
- **Liam's statement revisited**: "William is a fool."
  Since William is a sage, Liam's statement is false. Therefore, Liam must be a fool.
- **Emily's statement revisited**: "If Emily is a sage then Liam is a fool."
  Since Emily is a sage (from earlier), according to her statement, Liam must be a fool. This confirms our previous determination.
- **Owen's initial statement**: "William is a sage."
  Since Owen is a sage and William is a sage, Owen's statement is true. This is consistent.

**Therefore, summarizing:**

- Owen is a sage.
- Liam is a fool.
- Evelyn's statement was a contradiction, so Evelyn must be a fool.
- William is a sage.
- Sofia is a sage.
- Emily's statement was consistent, so Emily must be a sage.

$Owen$ is a sage, $Liam$ is a fool, $Evelyn$ is a fool, $William$ is a sage, $Sofia$ is a sage, and $Emily$ is a sage

---

To estimate Solution Multiplicity, we query the GPT-5-Think model on 200 randomly selected samples with the following prompt:

---

**Prompt for Querying Solution Multiplicity**

**[INST]**You are an expert mathematics educator and problem solver. Analyze the given mathematical problem and determine how many different solution approaches exist for it.

Please provide a comprehensive analysis that: 1. Identifies all distinct solution methods/approaches 2. Briefly explains what each approach involves 3. Counts the total number of different approaches

Mathematical Problem: problem

Please first explain what different solution approaches exist for this problem, then provide your final answer in the format: <ways> [number] </ways>

For example, if a problem has exactly 2 different solution methods, your response should end with: <ways> 2 </ways> **[/INST]**

---

*Table A6: Illustration of Experimental Result*

| Task | Knight and Knaves | Math | Countdown-3 | Countdown |
|---|---|---|---|---|
| **Solution Multiplicity** | 1.5 | 3.7 | 6.5 | 15.7 |
| **Pass@8 of GRPO** | 47.1 | 78.6 | 97.7 | 73.4 |
| **Pass@8 of GRPO** + Entropy bonus | 38.1 | 72.6 | 98.7 | 76.8 |
| **Entropy Effect for Pass@8** | -9.0% | -6.0% | +1.0% | +3.4% |

**Training details.**  The training setups of Math reasoning and Countdown task are identical to the main experiments as described in Appendix C.1 and C.3. For Countdown-3, we train the model for 160 steps. For GRPO with entropy bonus, we use a bonus coefficient of $\eta_+ = 0.05$. Other configurations are identical to those of the main experiment. For Knights-and-Knaves, we RL fine-tune the Qwen2.5-7B-Instruct Qwen Team (2024) model with LoRA adaptation (rank 256) (Hu et al., 2022) for 100 steps. We use a learning rate of $4 \times 10^{-5}$ and a batch size of 32, with 8 rollouts per sample.

**Result Analysis.**  The experimental results, presented in Table A6, reveal a direct correlation between Solution Multiplicity and the efficacy of entropy regularization. Specifically, as a task's Solution Multiplicity increases, so does the performance gain (Pass@8) of an entropy bonus over vanilla GRPO. This provides strong empirical support for our hypothesis: for tasks with a larger solution space, the benefits of enhanced diversity outweigh the potential trade-offs in single-solution correctness. These findings thus validate Solution Multiplicity as a practical metric for guiding the decision of whether to increase or decrease entropy for a given task.

# D  ADDITIONAL EXPERIMENTAL RESULTS FOR COUNTDOWN

In this section, we provide additional results for the Countdown task.

## D.1  ADDITIONAL EXPERIMENTS FOR ENTROPY COEFFICIENT

To provide a more comprehensive comparison, we analyze the performance of the entropy-based baselines across their full hyperparameter sweep. We compare DS-GRPO against GRPO with varying entropy bonus ($\eta_+$) and penalty ($\eta_-$) coefficients, with the results illustrated in Figure A1 (Top). The figure clearly demonstrates that DS-GRPO consistently outperforms the global entropy control methods across their entire range of tested hyperparameters for all values of $K$.

## D.2  EFFECTS OF KL COEFFICIENT AND OTHER FACTORS

Figure A1 (Bottom Left) reports results with varying sampling temperatures for both DS-GRPO and GRPO. Under the same temperature, DS-GRPO achieves consistently higher Pass@$K$.

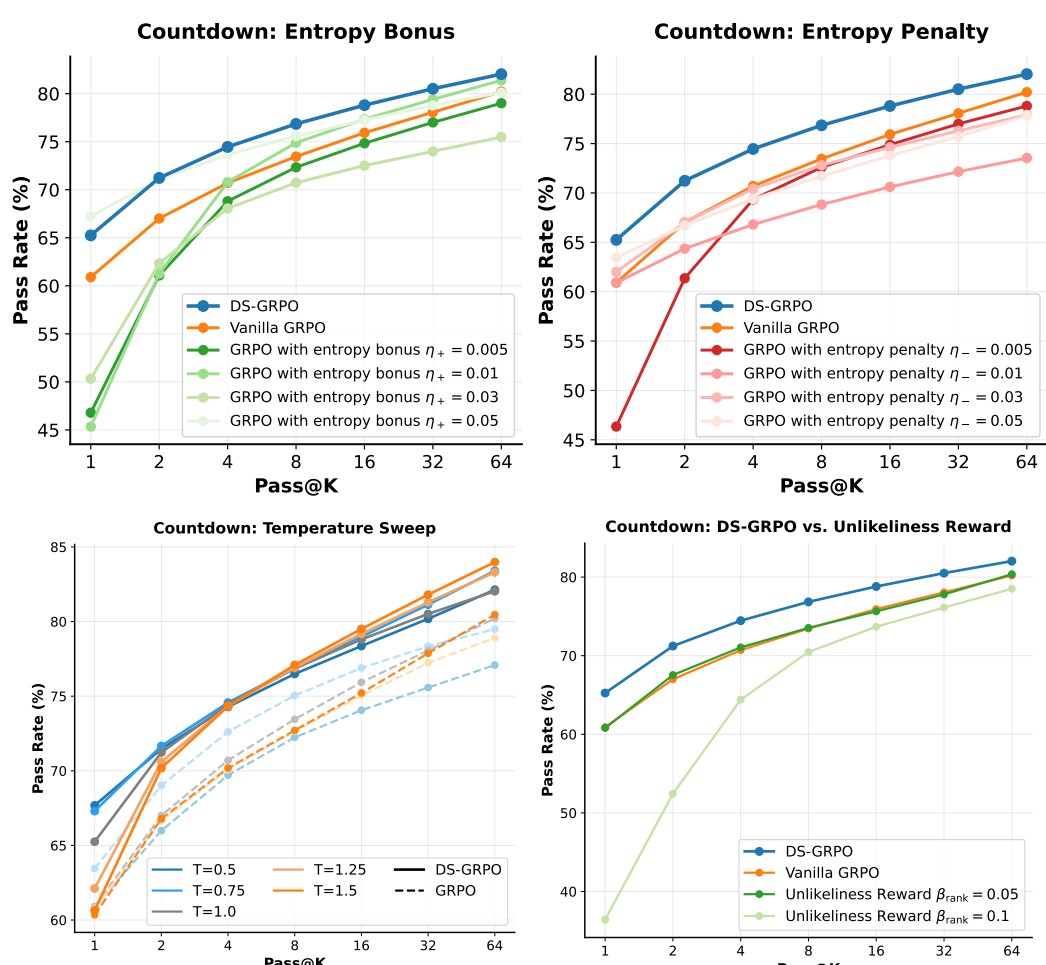

*Figure A1: Additional results on the Countdown task comparing DS-GRPO with various baselines. Top: Pass@K performance of DS-GRPO and GRPO under entropy bonus and entropy penalty. Bottom Left: Pass@K performance of DS-GRPO and GRPO across different sampling temperatures. Bottom Right: Pass@K performance of DS-GRPO and the Unlikeliness Reward method with varying coefficients.*

Figure A1 (Bottom Right) presents results from varying the unlikeliness reward coefficient $\beta_{\text{rank}} \in \{0.05, 0.1, 0.15, 0.2, 0.25, 0.3\}$ (He et al., 2025a). For $\beta_{\text{rank}} \geq 0.15$, training collapses and accuracy drops to 0, so we omit those results.

**Observed Experimental Performance:** Our experimental results show a differential effect:

- +Entropy (Bonus): Improves Pass@K on **Countdown**, but decreases Pass@K on **Math500**.
- −Entropy (Penalty): Improves Pass@K on **Math500**, but decreases Pass@1 and Pass@K on **Countdown**.

Our core explanation is rooted in the trade-off: the effect of +entropy is to enhance diversity but compromise correctness (P@1), while the effect of −entropy is to sharpen correctness but diminish diversity. The negative effect of increasing/decreasing diversity and correctness is task-dependent because the relative contribution of diversity and correctness to the final Pass@K score differs.

- +Entropy (Bonus): On **Countdown**, the positive effect of increasing diversity **outweighs** the negative effect of decreasing correctness, thus Pass@K improves. However, on **Math500**, the detrimental effect on correctness **outweighs** the benefit of increased diversity, causing Pass@K to decrease.

- −Entropy (Penalty): Conversely, for the entropy penalty, the effect of harming diversity **outweighs** the benefit of improving correctness on **Countdown**. Yet, on **Math500**, the improvement in correctness **outweighs** the harm to diversity, leading to an increase in Pass@K.

# E ADDITIONAL EXPERIMENTAL RESULTS FOR MATH REASONING EXPERIMENT

## E.1 ADDITIONAL EXPERIMENTAL RESULTS ON DS-GRPO VS GRPO

**Experimental Results.** Figure A4, which contains the full results for Section **??**, compares our proposed DS-GRPO against the vanilla GRPO baseline across three different base models and five mathematical reasoning benchmarks. The results consistently demonstrate that DS-GRPO outperforms vanilla GRPO across all tested models and datasets.

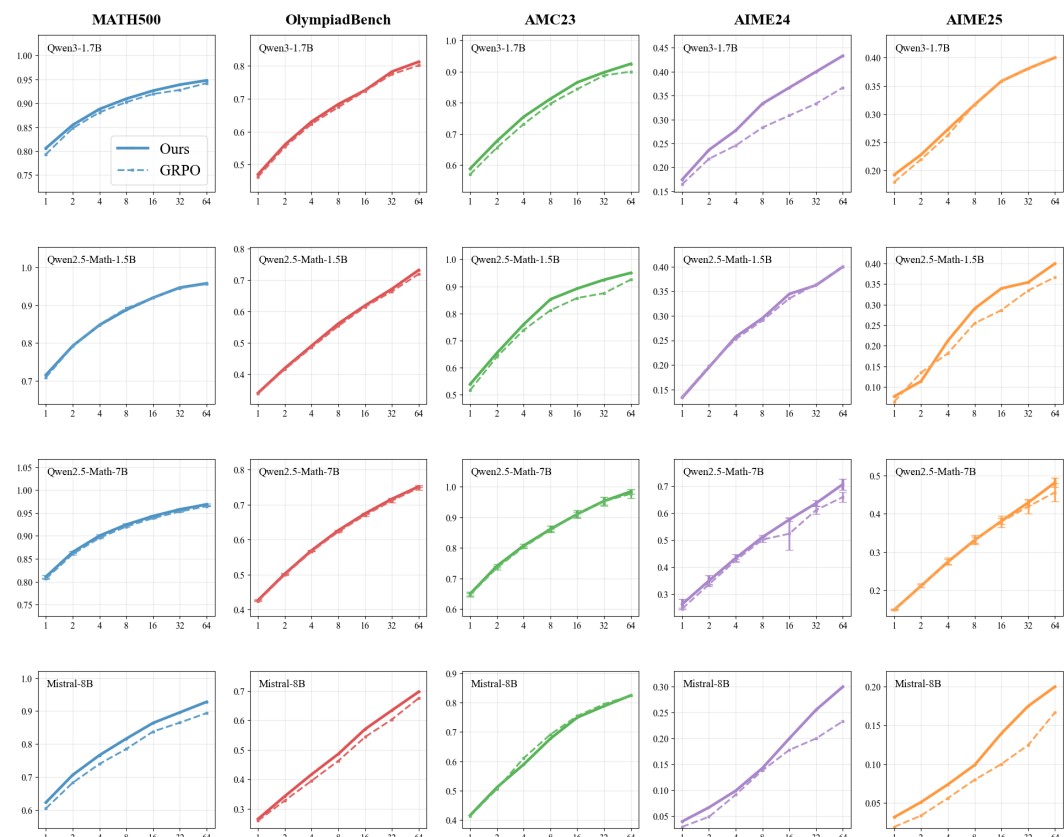

*Figure A2: Pass@K performance after reward modification, compared with vanilla GRPO. X-axis denotes K and y-axis denotes pass rates. Trained on the DAPO(Yu et al., 2025) and the MATH(Hendrycks et al., 2021) Dataset.*

## E.2 ADDITIONAL ABLATION STUDY FOR DS-GRPO

**Ablation Study Implementation.** To isolate the contribution of each component in our reward modification strategy, we conduct an ablation study. We compare the full DS-GRPO algorithm against two specialized variants: *DS-GRPO-Positive*, which only modifies the advantage for correct trajectories, and *DS-GRPO-Negative*, which only modifies the advantage for incorrect trajectories.

Their respective advantage modifications are defined as follows:

$$A_i^{\text{DS+}} = A_i - \gamma_p \ \log \pi_{\theta_{\text{old}}}(y_i \mid x), \quad \text{if } r_i = 1,$$
$$A_i^{\text{DS-}} = A_i + \gamma_n \ \log \pi_{\theta_{\text{old}}}(y_i \mid x), \quad \text{if } r_i \neq 1.$$

The DS-GRPO-Positive variant applies only the modification to correct trajectories ($A_i^{\text{DS+}}$), leaving the advantage for incorrect trajectories as the standard $A_i$. Conversely, the DS-GRPO-Negative variant applies only the modification to incorrect trajectories ($A_i^{\text{DS-}}$), leaving the advantage for correct

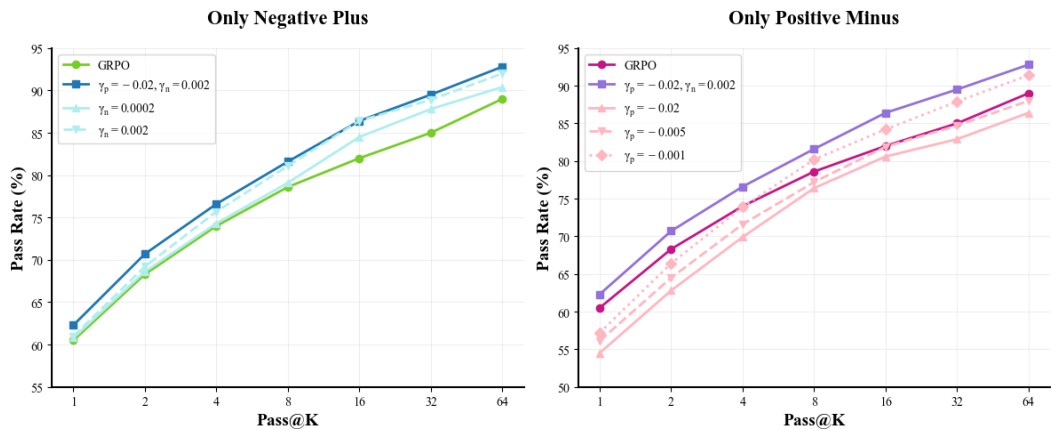

*Figure A3: Comparison on Different Hyperparameter of DS-GRPO.*

trajectories unchanged. **Result Analysis.** We present the results of our ablation study in Figure A3. The key findings are as follows:

- *DS-GRPO-Positive vs. Vanilla GRPO.* As shown in Figure A3, DS-GRPO-Positive outperforms vanilla GRPO, particularly for larger values of $K$. This demonstrates that modifying the reward for correct trajectories successfully mitigates the sharpening effect, providing empirical support for our intuition in Section 3.3 that penalizing high-probability correct solutions enhances diversity.

- *DS-GRPO-Negative vs. Vanilla GRPO.* The figure also shows that DS-GRPO-Negative consistently outperforms vanilla GRPO across all values of $K$. This indicates that modifying the reward for incorrect trajectories is effective at improving the model's overall correctness.

- *DS-GRPO vs. Its Components.* The full DS-GRPO algorithm demonstrates superior performance over both of its individual components (DS-GRPO-Positive and DS-GRPO-Negative) for all $K$. This highlights a clear synergy: the "Positive" component drives diversity, while the "Negative" component enhances correctness. Their combination in DS-GRPO achieves the best balance, validating our complete reward modification strategy as outlined in Section 3.3.

*Table A7: Pass@1 and Pass@64 sorted by $\gamma_p$ settings.*

| $\gamma_p$ | **0.02** | | | **0.03** | | | **0.04** | | |
|---|---|---|---|---|---|---|---|---|---|
| $\gamma_n$ | 0.005 | 0.020 | 0.040 | 0.005 | 0.010 | 0.030 | 0.005 | 0.020 | 0.040 |
| Pass@1 | 0.669 | 0.583 | 0.608 | 0.637 | 0.652 | 0.637 | 0.568 | 0.599 | 0.593 |
| Pass@64 | 0.841 | 0.734 | 0.824 | 0.807 | 0.820 | 0.807 | 0.820 | 0.782 | 0.814 |

# F  SUPPLEMENTARY EXPERIMENTS AND ANALYSIS

## F.1  PARAMETER SENSITIVITY

To evaluate the parameter sensitivity, we constructed a two-dimensional uniform grid over the identified intervals and evaluated the candidate combinations. Our results are shown as follows.

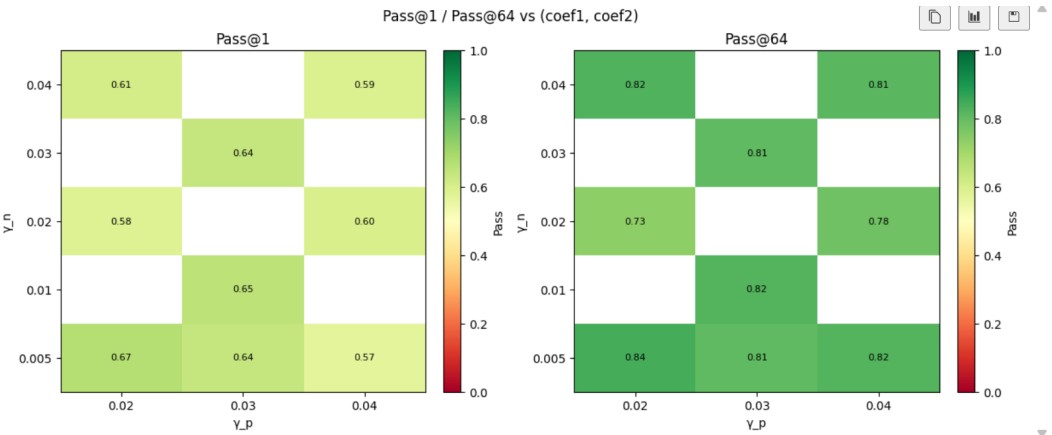

*Figure A4: Pass@K performance after reward modification, compared with vanilla GRPO. X-axis denotes K and y-axis denotes pass rates. Trained on the DAPO(Yu et al., 2025) and the MATH(Hendrycks et al., 2021) Dataset.*

We can observe that the performance of DS-GRPO on neither Pass@1 nor Pass@K is sensitive to parameters $\gamma_n$ and $\gamma_p$. Based on this observation, it is easy to find the best parameter combination with the method described below.

We employed a two-step procedure to select and optimize the hyperparameters $\gamma_n$ and $\gamma_p$:

1. **Step 1: Preliminary range identification (Coarse Search).** First, we determined a feasible interval for $\gamma_n$ and $\gamma_p$ through a coarse search. By fixing one parameter and varying the other, we observed that excessively large values for $\gamma_n$ or $\gamma_p$ led to training instability and significant performance degradation. Consequently, we established a rough search interval $[a_n, b_n] \times [a_p, b_p]$ within which the training remained stable.

2. **Step 2: Fine-grained selection via Grid Search** After defining the coarse intervals, we performed a fine-grained grid search to pinpoint the optimal combination. We constructed a two-dimensional uniform grid over the identified intervals and evaluated the candidate combinations $\left\{ \left( \gamma_n = \frac{i}{N(b_n - a_n)}, \gamma_p = \frac{j}{N(b_p - a_p)} \right) \right\}_{i,j \in [N]}$. We then selected the parameter set that achieved the best performance on the Pass@K and Pass@1 metrics. It is worth noting that the performance of DS-GRPO is relatively robust to hyperparameter variations within this effective range (please refer to the sensitivity analysis in the following part).

### F.2 REDUNDANCY ANALYSIS WITH ENTROPY REGULARIZATION.

We further investigate whether adding entropy regularization complements DS-GRPO or if its effects are redundant. Our hypothesis is that the exploration benefits of entropy regularization are implicitly captured by DS-GRPO. To verify this, we conducted experiments on the Countdown dataset using the Qwen2.5-3B-Instruct model. Initial results showed that adding entropy regularization to DS-GRPO yields improvements over vanilla GRPO. However, by removing entropy regularization and instead fine-tuning DS-GRPO hyperparameters (specifically, increasing $\gamma_p$ and decreasing $\gamma_n$), we achieved superior performance compared to the combined approach. This demonstrates that the benefits associated with entropy regularization can be effectively subsumed by optimizing DS-GRPO directly. The comparative results are presented in Figure A5.

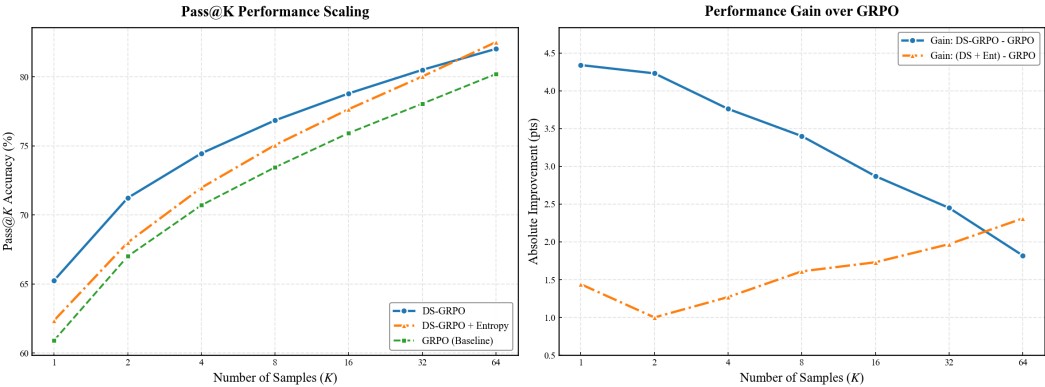

*Figure A5: Pass@K performance of DS-GRPO and DS-GRPO with entropy regularization*

### F.3 ADDITIONAL EXPERIMENT ON COMPARING DS-GRPO WITH CISPO

We compare DS-GRPO with CISPO (Chen et al., 2025a), as illustrated in Figure A6. The results demonstrate that DS-GRPO consistently achieves a higher Pass@K compared to CISPO across all datasets. All experiments were conducted using the Qwen2.5-Math-1.5B model.

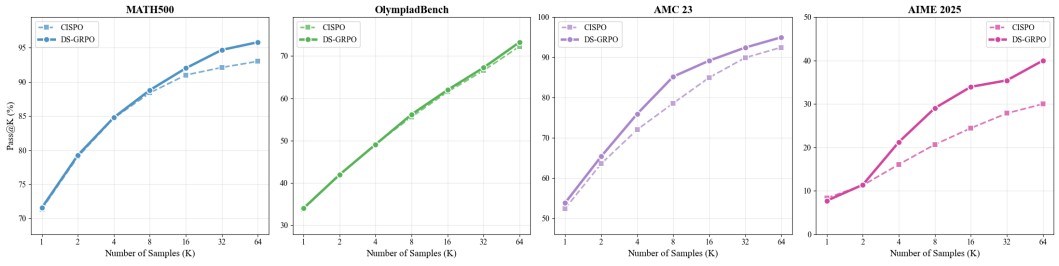

*Figure A6: Comparison on CISPO with DS-GRPO.*

### F.4 DIVERSITY CHANGES OF DS-GRPO COMPARING TO BASELINE MODEL

To provide stronger evidence that DS-GRPO truly mitigates entropy collapse, follow the convention of (Hochlehnert et al., 2025). We plot the Pass@K difference between DS-GRPO with base model over K in the following figure. We can see that the Pass@K difference between DS-GRPO with base model does not go down as K increases. This shows that DS-GRPO truly mitigates diversity collapse. The figure is shown in Fig A7.

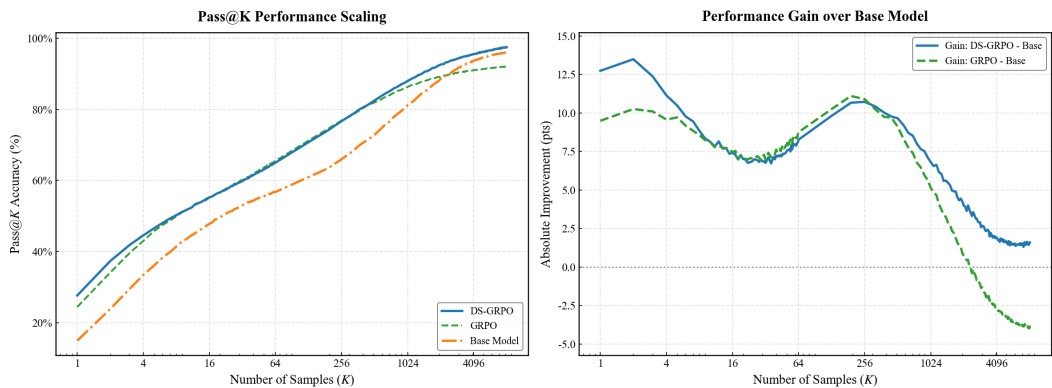

*Figure A7: Pass@K change of GRPO and DS-GRPO.*

### F.5 COMPARISON OF DIFFERENTIAL ENTROPY CONTROL WITH OTHER ENTROPY BASED METHODS

We extended our experiments to four mathematical reasoning datasets, comparing our method against Entropy Bonus, Entropy Penalty, and vanilla GRPO. The results are shown in Figure A8, our proposed method outperforms competing baselines in the vast majority of settings. The results highlight the performance gain of our Differential Entropy method over vanilla GRPO and other entropy-based variants. Notably, our method exhibits consistent performance gains across varying $K$, particularly demonstrating superior scaling capability at higher $K$ values (e.g., $K \geq 4$).

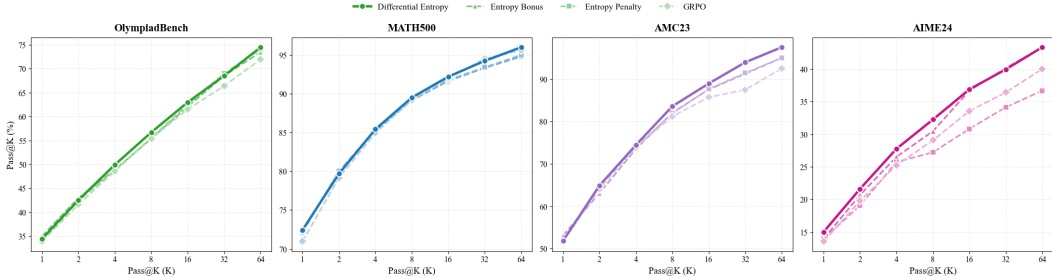

*Figure A8: Pass@K change of GRPO and DS-GRPO.*

### F.6 STATISTICAL ANALYSIS OF DS-GRPO IMPROVEMENTS

We further quantify the performance gains of DS-GRPO compared to the GRPO baseline. The improvements, averaged across experimental runs, are detailed in Table A8. The results demonstrate that DS-GRPO yields consistent and positive uplifts in Pass@K metrics across all evaluated datasets, validating the robustness of our method.

*Table A8: Average performance improvement of DS-GRPO over GRPO across different datasets. The values represent the percentage point increase in Pass@K.*

| Dataset | P@1 | P@2 | P@4 | P@8 | P@16 | P@32 | P@64 |
|---|---|---|---|---|---|---|---|
| MATH500 | +0.8% | +0.7% | +0.8% | +0.8% | +0.8% | +1.0% | +0.9% |
| AIME 2024 | +1.1% | +1.3% | +1.2% | +1.5% | +4.1% | +3.3% | +4.6% |
| AIME 2025 | +0.6% | +0.0% | +0.9% | +0.8% | +1.6% | +1.7% | +2.4% |
| OlympiadBench | +0.4% | +0.6% | +0.7% | +0.9% | +0.8% | +1.0% | +1.0% |
| AMC 2023 | +1.4% | +1.3% | +0.8% | +1.4% | +1.7% | +1.8% | +1.7% |

