# OpenReview forum: "Differential Smoothing Mitigates Sharpening and Improves LLM Reasoning"
_ICLR.cc/2026/Conference — Submitted to ICLR 2026_

### Official Review · Reviewer_jhoN · 2025-10-30

**Soundness:** 3
**Presentation:** 4
**Contribution:** 3
**Rating:** 8
**Confidence:** 3

**Summary:**

The authors tackle the problem of diversity collapse when finetuning LMs with reinforcement learning. The phenomenon is referred to as the sharpening effect and the authors show theoretically that this effect arises during RL finetuning of LMs.

To mitigate the sharpening effect in RL the authors propose a modification of the reward function—differential smoothing---to down-weight correct samples and upweight incorrect samples by the likelihood under the base model. The authors prove that this mitigates the sharpening effect in vanilla RL. In practice, the authors propose differential-smoothing GRPO or DS-GRPO which modifies the GRPO advantages with the corresponding likelihoods depending on the trajectory being correct or incorrect.

On Countdown and Math500 datasets the authors show empirically that this both improves pass@1—overall correctness—and pass@k—diversity. The authors also provide an explanation regarding how entropy penalties and entropy bonuses have both been shown to improve performance on distinct datasets by showing a relationship with the number of unique correct solutions; entropy bonuses work well with datasets which have a high number of unique correct solutions.

Additionally in the vanilla RL setting the authors show theoretically that DS-GRPO surpasses both vanilla GRPO and direct policy entropy maximization.

**Strengths:**

* Strong theoretical underpinning of the DS-GRPO method put forward.
* Two theoretical analyses:
    * Demonstrating the sharpening effect in RL
    * Showing that DS-GRPO surpass vanilla GRPO and direct entropy maximization in terms of correctness and trajectory diversity/
* Very interesting analysis showing that entropy penalties benefit the MATH500 dataset while an entropy bonus helps with the Countdown dataset. There is a correlation between the uplift in performance using an entropy bonus with the solution diversity of correct answers. So problems with a high solution diversity will benefit from an entropy bonus and problems with low solution diversity will not.

**Weaknesses:**

There are a few of weaknesses with the empirical presentation:
* Figure 3 in the main text has pass@k curves which are cherry picked from Figure A2 in the appendix. A more thorough presentation of the results would be to plot the % uplift in the performance using DS-GRPO over GRPO and average over model families for each dataset. This would give a notion of uncertainty as well and if the effect of DS-GRPO is statistically significant versus vanilla GRPO.
* In lines 387 to 390 the authors note that an entropy bonus improves performance over vanilla GRPO on one datasets but an entropy penalty then improves performance on a different dataset and that these trends are differing, however the difference between these two approaches is only very obvious on Math500 not on Countdown, Figure 4 (right). I think to properly draw this conclusion the authors should repeat the experiments with multiple seeds to make sure that this effect is not simply due to experimental noise, obviously I understand resource constraints though.

**Questions:**

Minor: small question regarding the theoretical analysis in Section 6. The use of $\sigma$ is usually used to define a standard deviation, but it is being used to define a variance with $\sigma_{DS}$?

Has this entropy collapse/sharpening effect been observed in vanilla RL prior to LMs? Or is this an LM specific phenomenon when using RL finetuning?

---

> ### Author Response · Authors · 2025-11-24
> **Rebuttal Part 1**
>
> We sincerely thank reviewer jhoN for the valuable comments and positive feedback on our work. Below, we would like to address each of your concerns and questions:
> ## Weakness 1: A thorough process of the result comparing GRPO and DS-GRPO
>
> >**Weakness:** A more thorough presentation of the results would be to plot the % uplift in the performance using DS-GRPO over GRPO and average over model families for each dataset.
>
> Thank you for these valuable suggestions and questions. In response, we have conducted additional experiments and performed a statistical analysis of the results. We explain the details as follows:
>
> * **We add an extra 7B model and multiple runs to verify the optimality of our method.**
>     We first implemented DS-GRPO on an additional 7B model, Qwen2.5-Math-7B. Furthermore, we conducted comparative experiments between DS-GRPO and vanilla GRPO over multiple runs (N=3) to confirm the superiority of our method. Our averaged results are presented in Table 1 below:
>
> * **Table 1: Averaged performance of GRPO and DS-GRPO across multiple datasets (Mean $\pm$ Std. Dev. over 3 runs).**
>
> | Dataset | Method | P@1 | P@2 | P@4 | P@8 | P@16 | P@32 | P@64 |
> | :--- | :--- | :--- | :--- | :--- | :--- | :--- | :--- | :--- |
> | **AIME 2024** | GRPO | $24.55 \\pm 0.31$ | $33.55 \pm 0.16$ | $42.43 \pm 0.73$ | $50.46 \pm 1.25$ | $55.97 \pm 7.42$ | $61.55 \pm 1.49$ | $65.67 \pm 2.03$ |
> | | **DS-GRPO** | $\mathbf{26.21 \pm 2.68}$ | $\mathbf{35.05 \pm 2.68}$ | $\mathbf{43.97 \pm 1.69}$ | $\mathbf{51.73 \pm 0.89}$ | $\mathbf{57.53 \pm 0.90}$ | $\mathbf{63.76 \pm 1.00}$ | $\mathbf{70.58 \pm 2.47}$ |
> | **AIME 2025** | GRPO | $14.99 \pm 0.07$ | $21.33 \pm 0.28$ | $27.59 \pm 0.86$ | $33.26 \pm 1.46$ | $37.91 \pm 1.94$ | $41.87 \pm 1.76$ | $45.57 \pm 2.83$ |
> | | **DS-GRPO** | $\mathbf{14.95 \pm 0.25}$ | $\mathbf{21.17 \pm 0.10}$ | $\mathbf{27.42 \pm 0.18}$ | $\mathbf{33.45 \pm 0.83}$ | $\mathbf{38.25 \pm 0.20}$ | $\mathbf{42.92 \pm 1.51}$ | $\mathbf{48.59 \pm 1.51}$ |
> | **AMC 23** | GRPO | $64.69 \pm 0.77$ | $73.95 \pm 0.63$ | $80.64 \pm 0.58$ | $85.87 \pm 0.40$ | $90.69 \pm 1.00$ | $95.58 \pm 1.70$ | $97.57 \pm 1.00$ |
> | | **DS-GRPO** | $\mathbf{65.00 \pm 0.24}$ | $\mathbf{74.44 \pm 0.28}$ | $\mathbf{80.93 \pm 0.78}$ | $\mathbf{86.50 \pm 1.00}$ | $\mathbf{91.41 \pm 1.53}$ | $\mathbf{97.10 \pm 1.43}$ | $\mathbf{98.58 \pm 0.86}$ |
> | **MATH 500** | GRPO | $80.65 \pm 0.19$ | $86.02 \pm 0.20$ | $89.56 \pm 0.20$ | $92.02 \pm 0.17$ | $93.86 \pm 0.13$ | $95.31 \pm 0.06$ | $96.46 \pm 0.08$ |
> | | **DS-GRPO** | $\mathbf{81.09 \pm 0.33}$ | $\mathbf{86.47 \pm 0.23}$ | $\mathbf{90.03 \pm 0.26}$ | $\mathbf{92.46 \pm 0.29}$ | $\mathbf{94.31 \pm 0.24}$ | $\mathbf{95.78 \pm 0.22}$ | $\mathbf{96.87 \pm 0.20}$ |
> | **OlympiadBench** | GRPO | $42.39 \pm 0.10$ | $49.93 \pm 0.09$ | $56.57 \pm 0.10$ | $62.23 \pm 0.20$ | $67.00 \pm 0.47$ | $71.45 \pm 0.15$ | $75.09 \pm 0.22$ |
> | | **DS-GRPO** | $\mathbf{42.68 \pm 0.20}$ | $\mathbf{50.21 \pm 0.19}$ | $\mathbf{56.90 \pm 0.14}$ | $\mathbf{62.64 \pm 0.07}$ | $\mathbf{67.46 \pm 0.18}$ | $\mathbf{71.61 \pm 0.18}$ | $\mathbf{75.23 \pm 0.34}$ |
>
> ---
>
> * **The improvement of DS-GRPO over GRPO is statistically significant.**
>     Following your suggestion, we further analyzed the performance uplift of DS-GRPO over GRPO and averaged the improvement across all relevant datasets. The average performance improvements are presented in Table 2 below. This analysis confirms that the improvement of DS-GRPO over vanilla GRPO on Pass@K for all $K$ is statistically significant.
>
> * **Table 2: Average performance improvement of DS-GRPO over GRPO across different datasets and models.**
>
> | Dataset | P@1 | P@2 | P@4 | P@8 | P@16 | P@32 | P@64 |
> | :--- | :--- | :--- | :--- | :--- | :--- | :--- | :--- |
> | MATH500 | +0.8\% | +0.7\% | +0.8\% | +0.8\% | +0.8\% | +1.0\% | +0.9\% |
> | AIME 2024 | +1.1\% | +1.3\% | +1.2\% | +1.5\% | +4.1\% | +3.3\% | +4.6\% |
> | AIME 2025 | +0.6\% | +0.0\% | +0.9\% | +0.8\% | +1.6\% | +1.7\% | +2.4\% |
> | OlympiadBench | +0.4\% | +0.6\% | +0.7\% | +0.9\% | +0.8\% | +1.0\% | +1.0\% |
> | AMC 23 | +1.4\% | +1.3\% | +0.8\% | +1.4\% | +1.7\% | +1.8\% | +1.7\% |

---

> ### Author Response · Authors · 2025-11-24
> **Rebuttal Part 2**
>
> ## Weakness 2: Different pattern of entropy in countdown and Math500
>
> >**Weakness:** In lines 387 to 390 the authors note that an entropy bonus improves performance over vanilla GRPO on one datasets but an entropy penalty then improves performance on a different dataset and that these trends are differing, however the difference between these two approaches is only very obvious on Math500 not on Countdown, Figure 4 (right).
>
> Thank you for raising this insightful question. We believe this point deserves further clarification and discussion.
>
> * **Our primary objective is to compare the effect of a fixed method on different tasks.**
>   Our aim is not to compare the effects of the $+\text{entropy}$ (bonus) and $-\text{entropy}$ (penalty) terms on a single task, but rather to analyze how a fixed regularization method impacts performance across various tasks. We seek to demonstrate and explain that the effect of a given method varies significantly depending on the task's characteristics.
>
> * **Observed Experimental Performance:**
>   Our experimental results show a differential effect:
>     * **$+\text{Entropy}$ (Bonus):** Improves Pass@K on **Countdown**, but decreases Pass@K on **Math500**.
>     * **$-\text{Entropy}$ (Penalty):** Improves Pass@1 and Pass@K on **Math500**, but decreases Pass@1 and Pass@K on **Countdown**.
>
> * **Underlying Mechanism of Entropy Regulation:**
>   Our core explanation is rooted in the trade-off: the effect of $+\text{entropy}$ is to enhance diversity but compromise correctness (P@1), while the effect of $-\text{entropy}$ is to sharpen correctness but diminish diversity.
>
> * **Explanation for Task-Specific Variation:**
>   The net effect of increasing/decreasing diversity and correctness is task-dependent because the relative contribution of diversity and correctness to the final Pass@K score differs.
>     * **$+\text{Entropy}$ (Bonus):** On **Countdown**, the positive effect of increasing diversity **outweighs** the negative effect of decreasing correctness, thus Pass@K improves. However, on **Math500**, the detrimental effect on correctness **outweighs** the benefit of increased diversity, causing Pass@K to decrease.
>     * **$-\text{Entropy}$ (Penalty):** Conversely, for the entropy penalty, the effect of harming diversity **outweighs** the benefit of improving correctness on **Countdown**. Yet, on **Math500**, the improvement in correctness **outweighs** the harm to diversity, leading to an increase in Pass@K.
>
> * **Supplemental Verification with Multiple Seeds:**
>   To further confirm our conclusion and analysis, we conducted supplementary experiments with multiple seeds. The results, shown in Table 3 below, clearly support our analysis: on the Countdown task, $+\\text{entropy}$ increases Pass@K while $-\\text{entropy}$ decreases it.
>
> * **Table 3: Pass@k results ($k = 1,2,4,8,16,32,64$) of vanilla GRPO and GRPO with entropy bonus or penalty on the Countdown task (Mean $\pm$ Std. Dev.).**
>
> | Method | Pass@1 | Pass@2 | Pass@4 | Pass@8 | Pass@16 | Pass@32 | Pass@64 |
> | :--- | :--- | :--- | :--- | :--- | :--- | :--- | :--- |
> | GRPO | $0.610 \pm 0.0016$ | $0.671 \pm 0.0009$ | $0.708 \pm 0.0010$ | $0.735 \pm 0.0013$ | $0.759 \pm 0.0001$ | $0.780 \pm 0.0007$ | $0.802 \pm 0.0002$ |
> | GRPO with entropy penalty | $0.637 \pm 0.0013$ | $0.684 \pm 0.0126$ | $0.714 \pm 0.0181$ | $0.737 \pm 0.0186$ | $0.757 \pm 0.0188$ | $0.776 \pm 0.0201$ | $0.794 \pm 0.0214$ |
> | GRPO with entropy bonus | $0.548 \pm 0.1339$ | $0.652 \pm 0.0557$ | $0.716 \pm 0.0113$ | $0.748 \pm 0.0009$ | $0.770 \pm 0.0049$ | $0.789 \pm 0.0075$ | $0.807 \pm 0.0092$ |
>
> Thanks for your great suggestion and comment. We have added this clarification to the revised version of our paper.

---

> ### Author Response · Authors · 2025-11-24
> **Rebuttal Part 3**
>
> ## Question 1: Notation for normalized Variance.
>
> >**Question:** The use of $\sigma$ is usually used to define a standard deviation, but it is being used to define a variance with $\sigma_{DS}$?
>
> Thanks for raising the questions. We apologize for the typos and the confusion regarding the notations in the original submitted paper.
>
> Our clarification is as follows:
>
> * **We use $\sigma$ to denote the normalized measure of dispersion (relative standard deviation) of the policy's distribution on correct trajectories.**
>     We define $\sigma_{\text{DS}}$ as:
>
> $$
> \sigma_{\text{DS}}=\frac{\sum_{\tau\in\mathcal{C}}\pi_{\text{DS}}(\tau)^2-\left[\sum_{\tau\in\mathcal{C}}\pi_{\text{DS}}(\tau)\right]^2}{\left[\sum_{\tau\in\mathcal{C}}\pi_{\text{DS}}(\tau)\right]^2}
> $$
>
> This metric measures the relative dispersion of the policy's probability mass on correct trajectories. We use the $\sigma$ notation because it fundamentally represents a measure of relative standard deviation for the policy's distribution.
>
> We apologize for the typos and confusion. We have corrected these issues in the revised version of our work.
> ## Question 2: Entropy collapse in non-LM RL
>
> >**Question 2:** Has this entropy collapse/sharpening effect been observed in vanilla RL prior to LMs? Or is this an LM specific phenomenon when using RL finetuning?
>
>
>
>
> Thank you for raising this valuable question. We provide our explanation below:
>
> * **The sharpening effect is well-documented in general RL literature, predating LLMs.**
>   Diversity collapse is not unique to language models; it has been extensively observed in broader reinforcement learning settings ([1][6][7][8][9]). For instance, in traditional RL domains such as 2D Gridworlds or Atari 2600 [1], agents exhibit a strong bias toward learning policies and actions that they are initially more confident in or that are easier to access during early exploration.
>
> * **The underlying mechanism of sharpening in traditional RL parallels that in LLMs.**
>   The fundamental cause of sharpening in traditional RL aligns with what we observe in LLMs. Specifically, states or actions that have a higher initial probability of being visited are explored more frequently. Consequently, even if a certain state yields a higher reward, it is less likely to be discovered and reinforced if its initial reachability is low. This "rich-get-richer" dynamic in exploration distribution drives the sharpening effect in both domains.
>
> * **However, the sharpening effect is notably more severe in LLM reinforcement learning.**
>   We specifically focus on RL with Verifiable Rewards (RLVR) for LLMs because the sharpening effect here is particularly acute compared to traditional settings. Unlike many control and robotics tasks where dense process rewards are often available along the trajectory, RL for LLMs typically relies on outcome-only rewards. Optimization focuses almost entirely on a sparse signal at the final token. As shown in recent studies ([4][5]), this setup strongly amplifies mode-seeking behavior ([2][3]), making diversity collapse a more critical issue in LLMs than in environments with denser feedback signals.
>
> We have added these detailed explanations and references to the revised version of our paper.
>
> [1] Diversity-Driven Exploration Strategy for Deep Reinforcement Learning
>
> [2] Pinpointing Crucial Steps: Attribution-Based Credit Assignment for Verifiable Reinforcement Learning
>
> [3] Exploration in Deep Reinforcement Learning: A Survey
>
> [4] Assessing Diversity Collapse in Reasoning
>
> [5] UNDERSTANDING THE EFFECTS OF RLHF ON LLM GENERALISATION AND DIVERSITY
>
> [6] Acquiring Diverse Robot Skills via Maximum Entropy Deep Reinforcement Learning
>
> [7] Proximal Policy Optimization Algorithms
>
> [8] Visuomotor Policy Learning via Action Diffusion
>
> [9] Robot Manipulation Learning Using Generative Adversarial Imitation Learning

---

### Official Review · Reviewer_17fN · 2025-10-31

**Soundness:** 3
**Presentation:** 3
**Contribution:** 3
**Rating:** 4
**Confidence:** 3

**Summary:**

This paper provides a formal framework for the sharpening effect observed after reinforcement learning (RL) on Large Language Models (LLM) that leads to increased correctness but also reduced output diversity. Based on that, the authors introduce a new method called "differential smoothing" which aims to improve both, correctness and diversity. Further the authors formally prove that differential smoothing improves diversity.

**Strengths:**

- Rigorous theoretical framework: The paper provides a formal and intuitive theoretical analysis of the sharpening effect
- The method is tested on multiple reasoning benchmarks (MATH500, AIME24/25, OlympiadBench, AMC23, Countdown) and several model families (Qwen2.5, Qwen3, Ministral-8B)
- The paper introduces an improved entropy-based method as comparison

**Weaknesses:**

- The paper would benefit from a comparison of DS-GRPO with more recent RL methods such as CISPO[1]. This would clarify how the proposed approach performs relative to the current state of the art.
- The paper reports absolute diversity measures, but typically diversity collapse is assessed relative to the base model. Showing how diversity changes compared to the pre-training baseline (e.g. [1], Figure 12) would provide stronger evidence that DS-GRPO truly mitigates entropy collapse.
- Since the proposed method introduces a new form of entropy control, it would be valuable to compare it directly with other entropy-based methods to better isolate the specific contribution of the “differential” mechanism.
- All experiments are conducted on smaller models (<8B), which are known to behave differently under GRPO training. While this limitation is understandable given academic resource constraints, it would be interesting if the observed improvements hold for larger-scale models (e.g., 32B+), where RL dynamics may be different.

1: MiniMax-M1: Scaling Test-Time Compute Efficiently with Lightning Attention
2: A Sober Look at Progress in Language Model Reasoning

**Questions:**

On several benchmarks, the improvements of DS-GRPO over vanilla GRPO appear relatively small which raises the question if the suggested mechanism is actually the main reason for diversity collapse in RL fine-tuning. Maybe the authors could elaborate a bit more. Is the diversity after DS-GRPO training almost the same as for the base model?

---

> ### Author Response · Authors · 2025-11-24
> **Rebuttal Part 1**
>
> We thank Reviewer 17fN for their constructive comments and feedback on our work. We appreciate the time taken to evaluate our paper and address the specific concerns and questions below.
> ## Weakness 1: Comparison of DS-GRPO with CISPO see (section F.3 for figures)
> >**Weakness 1:** The paper would benefit from a comparison of DS-GRPO with more recent RL methods such as CISPO[1].
>
>
> **Our proposed DS-GRPO outperform CISPO.**
>
> Thank you for raising this valuable question. In response, we have conducted a supplementary experiment comparing DS-GRPO with CISPO [1] on mathematical reasoning tasks using the Qwen2.5-Math-1.5B model. The results are presented below. We observe that our method (DS-GRPO) consistently achieves higher Pass@K scores across all values of $K$ and on all evaluated datasets. We also find that the gains are larger for higher $K$ showing that DS-GRPO is in fact better at preserving diversity than CISPO. We have cited [1] and included a detailed discussion regarding CISPO in the revised version of our paper.
>
> **Table: Performance comparison of our method (DS-GRPO) versus the baseline (CISPO). Improvements are calculated as (DS-GRPO - CISPO).**
>
> | Dataset | Model | P@1 | P@2 | P@4 | P@8 | P@16 | P@32 | P@64 |
> | :--- | :--- | :--- | :--- | :--- | :--- | :--- | :--- | :--- |
> | **MATH500** | CISPO | 71.38 | 79.04 | 84.70 | 88.39 | 91.00 | 92.11 | 93.00 |
> | | **DS-GRPO (Ours)** | 71.58 | 79.27 | 84.79 | 88.79 | 92.00 | 94.68 | 95.80 |
> | | *Improvement* | **+0.20** | **+0.23** | **+0.09** | **+0.40** | **+1.00** | **+2.57** | **+2.80** |
> | **OlympiadBench** | CISPO | 33.83 | 41.93 | 49.22 | 55.59 | 61.51 | 66.54 | 72.25 |
> | | **DS-GRPO (Ours)** | 34.04 | 41.95 | 49.09 | 56.18 | 61.96 | 67.23 | 73.24 |
> | | *Improvement* | **+0.21** | **+0.02** | -0.13 | **+0.59** | **+0.45** | **+0.69** | **+0.99** |
> | **AMC 23** | CISPO | 52.50 | 63.62 | 72.08 | 78.63 | 85.00 | 89.94 | 92.50 |
> | | **DS-GRPO (Ours)** | 53.91 | 65.42 | 75.99 | 85.24 | 89.21 | 92.45 | 95.00 |
> | | *Improvement* | **+1.41** | **+1.80** | **+3.91** | **+6.61** | **+4.21** | **+2.51** | **+2.50** |
> | **AIME 2025** | CISPO | 8.33 | 11.27 | 16.07 | 20.66 | 24.43 | 27.85 | 30.00 |
> | | **DS-GRPO (Ours)** | 7.71 | 11.39 | 21.23 | 29.05 | 33.95 | 35.44 | 40.00 |
> | | *Improvement* | -0.62 | **+0.12** | **+5.16** | **+8.39** | **+9.52** | **+7.59** | **+10.00** |
>
> [1]: MiniMax-M1: Scaling Test-Time Compute Efficiently with Lightning Attention

---

> ### Author Response · Authors · 2025-11-24
> **Rebuttal Part 2**
>
> ## Weakness 2: Diversity before and after DS-GRPO (see section F.4 for figure)
>
>
> >**Weakness:** Showing how diversity changes compared to the pre-training baseline (e.g. [1], Figure 12) would provide stronger evidence that DS-GRPO truly mitigates entropy collapse.
>
>
> **Comparative analysis against the pre-training baseline confirms that our method effectively mitigates diversity collapse.**
>
> We thank the reviewer for raising this critical point. In response, we conducted an experiment to analyze diversity shifts relative to the pre-training baseline, strictly adhering to the methodology described in [2]. Specifically, we evaluated Pass@$K$ across a comprehensive range of $K=1, 2, \ldots, 8192$, consistent with the analysis presented in Figure 12 of [2]. Representative results are provided below, while full data and visualizations have been included in the Appendix of the revised manuscript.
>
> Comparing DS-GRPO with the base model, we observe that DS-GRPO consistently maintains higher performance, supporting the conclusion that **diversity collapse is effectively prevented**. In contrast, the standard GRPO exhibits clear signs of diversity collapse, evidenced by its Pass@$K$ falling below that of the base model at high $K$.
>
> **Table 1: Performance gain ($\Delta$) relative to the Base Model (in percentage points). Note that GRPO underperforms the base model ($\Delta < 0$) at large K values, while DS-GRPO maintains a positive gain consistently.**
>
> | Method ($\Delta$) | K=1 | K=4 | K=16 | K=64 | K=128 | K=512 | K=1024 | K=2048 | K=3072 | K=4096 | K=6144 | K=8192 |
> | :--- | :---: | :---: | :---: | :---: | :---: | :---: | :---: | :---: | :---: | :---: | :---: | :---: |
> | GRPO - Base | +9.49 | +9.57 | +7.45 | +8.75 | +10.25 | +9.11 | +5.14 | +0.71 | **-1.44** | **-2.58** | **-3.68** | **-3.80** |
> | DS-GRPO - Base | +12.73 | +11.14 | +7.41 | +8.27 | +9.81 | +9.65 | +6.81 | +3.98 | +2.67 | +1.91 | +1.40 | +1.60 |

---

> ### Author Response · Authors · 2025-11-24
> **Rebuttal Part 3**
>
> ## Weakness 3: Compare differential entropy with entropy based method (See section F.5 for figure)
>
> >**Weakness:** Since the proposed method introduces a new form of entropy control, it would be valuable to compare it directly with other entropy-based methods to better isolate the specific contribution of the “differential” mechanism.
>
>
> **Our method demonstrates robust improvement over alternative entropy-based baselines.**
>
> We thank the reviewer for this valuable suggestion. We agree that comparing our approach with other entropy-based methods significantly strengthens the evaluation. Accordingly, we extended our experiments to four mathematical reasoning datasets, comparing our method against Entropy Bonus, Entropy Penalty, and vanilla GRPO. As detailed in the following table, our proposed method outperforms competing baselines in the vast majority of settings. The results highlight the performance gain of our Differential Entropy method over vanilla GRPO and other entropy-based variants. Notably, our method exhibits consistent performance gains across varying $K$, particularly demonstrating superior scaling capability at higher $K$ values (e.g., $K \ge 4$).
>
> **Table 1: Performance Gain of Differential Entropy over Baselines ($\Delta = \text{Diff. Entropy} - \text{Baseline}$) at varying Pass@K.**
>
> | Dataset | vs. Baseline | K=1 | K=2 | K=4 | K=8 | K=16 | K=32 | K=64 |
> | :--- | :--- | :---: | :---: | :---: | :---: | :---: | :---: | :---: |
> | **OlympiadBench** | Entropy Bonus | -0.58 | -0.44 | **+1.08** | **+1.32** | **+0.89** | 0.00 | **+1.02** |
> | | Entropy Penalty | **+0.55** | **+0.11** | **+1.13** | **+1.16** | **+0.49** | -0.46 | **+0.42** |
> | | GRPO | **+0.55** | **+0.93** | **+1.35** | **+1.29** | **+1.48** | **+2.11** | **+2.53** |
> | **MATH500** | Entropy Bonus | **+0.27** | -0.29 | **+0.18** | **+0.37** | **+0.55** | **+0.91** | **+1.20** |
> | | Entropy Penalty | **+0.09** | -0.33 | **+0.01** | **+0.25** | **+0.40** | **+0.82** | **+1.00** |
> | | GRPO | **+1.48** | **+0.51** | **+0.48** | **+0.32** | **+0.20** | -0.28 | **+0.40** |
> | **AMC23** | Entropy Bonus | -1.27 | **+1.80** | **+0.93** | **+1.53** | **+1.41** | **+2.70** | **+2.50** |
> | | Entropy Penalty | **+0.16** | **+0.06** | **+0.51** | **+1.65** | **+1.31** | **+2.52** | **+2.50** |
> | | GRPO | **+0.15** | **+0.87** | **+0.55** | **+2.39** | **+3.24** | **+6.45** | **+5.00** |
> | **AIME24** | Entropy Bonus | **+1.05** | **+0.94** | **+1.23** | **+1.79** | **+0.20** | -0.20 | 0.00 |
> | | Entropy Penalty | **+1.05** | **+2.56** | **+2.03** | **+5.03** | **+6.05** | **+5.81** | **+6.66** |
> | | GRPO | **+1.46** | **+1.83** | **+2.51** | **+3.19** | **+3.33** | **+3.53** | **+3.33** |
>
> ## Weakness 4: All experiments are conducted on smaller models
>
> **We add another 7B model experiment to verify our results**
> We appreciate the reviewer's understanding regarding our computational resource constraints. Regrettably, the 7B parameter scale represents the upper bound of our current experimental capacity. However, to further validate the robustness of our method, we have conducted additional experiments on an extra 7B model (Qwen2.5-Math-7B). The results are presented below:
>
> **Table: Performance improvement ($\Delta$) of DS-GRPO over GRPO on Qwen2.5-Math-7B.** Values are calculated as (DS-GRPO - GRPO) * 100.
>
> | Dataset | K=1 | K=2 | K=4 | K=8 | K=16 | K=32 | K=64 |
> | :--- | :---: | :---: | :---: | :---: | :---: | :---: | :---: |
> | MATH500 | +0.47 | +0.47 | +0.54 | +0.55 | +0.58 | +0.67 | +0.70 |
> | OlympiadBench | +0.52 | +0.52 | +0.47 | +0.20 | -0.09 | 0.00 | +0.35 |
> | AMC 23 | +2.19 | +1.38 | +2.08 | +4.04 | +3.49 | +4.95 | +2.50 |
> | AIME 24 | -0.73 | -1.03 | -1.20 | -1.15 | -0.67 | +0.42 | +2.10 |
> | AIME 25 | -0.28 | +0.15 | +0.32 | +0.25 | +0.79 | +2.26 | +3.87 |
>
>
>
> ## Question 1: Is the diversity after DS-GRPO training almost the same as for the base model?
>
> >**Question:** On several benchmarks, the improvements of DS-GRPO over vanilla GRPO appear relatively small which raises the question if the suggested mechanism is actually the main reason for diversity collapse in RL fine-tuning. Maybe the authors could elaborate a bit more. Is the diversity after DS-GRPO training almost the same as for the base model?
>
> As detailed in our response to Weakness 2, we conducted an experiment to analyze diversity shifts relative to the pre-training baseline. Our results show that while vanilla GRPO exhibits reduced diversity compared to the pre-trained model, DS-GRPO successfully recovers the original diversity levels. This empirically validates that our proposed mechanism effectively addresses the root cause of diversity collapse in RL fine-tuning.

---

> ### Comment · Reviewer_17fN · 2025-11-25
> **Response to Authors' Rebuttal**
>
> I thank the authors for addressing my questions and conducting additional experiments. They added a comparison with CISPO, demonstrating that their method outperforms this baseline and establishing its relevance. They further show that their differential entropy approach improves over existing methods, and, most importantly, they directly compare the diversity of the trained model with that of the base model, clearly demonstrating that their method prevents diversity collapse. Therefore, I will increase my score.

---

### Official Review · Reviewer_fC3U · 2025-10-31

**Soundness:** 3
**Presentation:** 3
**Contribution:** 3
**Rating:** 6
**Confidence:** 2

**Summary:**

the paper addresses the reasoning sharpening problem in RL fine-tuning of LLMs. RL fine-tuning often lets models over-optimize and hence lose diversity. The paper proposes a differential smoothing mechanism (DS-GRPO algorithm) to counteract this issue. The papers shows that this change can help keep the expected reward unbiased while lowering variance and preventing entropy collapse.

**Strengths:**

- The sharpening and lack of diversity is an important problem studied
- The paper provides a theoretically justified method for addressing the sharpening issue of RL fine-tuning, and the solution is simple to adapt for existing RL algorithms
- The paper provides solid empirical evidence across various reasoning benchmarks

**Weaknesses:**

I'm not an expert on the reasoning datasets, but it seems that the pass rate changes from the original GRPO and the proposed DS-GRPO are consistent but also modest.

**Questions:**

Is the improvement additive to entropy regularization if we combine both, or are the effects redundant?

---

> ### Author Response · Authors · 2025-11-24
> **Rebuttal Part 1**
>
> We thank the reviewer for the positive assessment. In our response, we have contextualized the performance gains achieved by DS-GRPO, which we believe to be substantial. We also address the potential for combining DS-GRPO with entropy control mechanisms. We hope these clarifications further strengthen your favorable evaluation of our work.
>
>
> ### Weakness 1: pass rate changes from GRPO and DS-GRPO are modest
>
> >**Weakness:** it seems that the pass rate changes from the original GRPO and the proposed DS-GRPO are consistent but also modest.
>
> * **We respectfully believe that the gains offered by DS-GRPO are in fact significant, in the context of gains typically seen in the specific settings we consider in our work of LLM reasoning via RL.**
>
> **Having contextualized the performance gains, we maintain that the improvements achieved by DS-GRPO are significant and meaningful for the field.**
>
>
> Our method achieves up to $6.7\%$ improvement with Mistral 8B on AIME24 on Pass@K comparing with baseline method (vanilla GRPO) . Notably, our result has one of the highest improvements over vanilla algorithm comparing with other recent works that also aim at improving LLM generation diversity. We list some of the recent works as follows. These related works evaluate their methods on mathematical reasoning tasks under settings comparable to ours, with several utilizing identical datasets and model sizes ranging from 1.5B to 7B. We can observe that we achieve the highest improvement on Pass@K. Consequently, we consider the observed gain of approximately 6.7% on mathematical reasoning benchmarks to be a significant and substantial advancement relative to the state-of-the-art.
>
>
>
>
> | Metric | [1] | [2] | [3] | [4] | [5] | Ours |
> | :--- | :---: | :---: | :---: | :---: | :---: | :---: |
> | $\uparrow$ Pass@K ($K\leq 64$) | 1.8%-3.0% | 3.9%-4.2% | 5.1% | 5.0%-6.0% | 0.3%-3.2% | **6.7%** |
> * **Our method outperforms other methods that aim to improve LLM generation diversity.**
>     We also compare our method with some of recently proposed methods which aim to improve LLM generation diversity. We find that our method outperforms other similar aimed methods on Pass@K for all $K$ and all different tasks. We show that although some recently proposed methods perform well on their specifically designed datasets, it does not perform well on real-world reasoning tasks like math reasoning. Our result is shown in the experiment part of our paper. We also represent it as follows:
>
>     | Method | P@1 | P@2 | P@4 | P@8 | P@16 | P@32 | P@64 |
>     | :--- | :---: | :---: | :---: | :---: | :---: | :---: | :---: |
>     | Entropy bonus | 51.3 | 60.1 | 67.4 | 72.6 | 78.1 | 81.4 | 84.8 |
>     | Unlikeliness Reward | 57.5 | 66.0 | 72.4 | 78.0 | 81.1 | 85.0 | 88.4 |
>     | GR-PKPO | 59.8 | 68.2 | 74.4 | 79.3 | 82.9 | 86.2 | 89.2 |
>     | GRPO | 60.5 | 68.3 | 74.0 | 78.6 | 82.0 | 85.4 | 88.2 |
>     | Entropy penalty | 61.4 | 69.9 | 76.0 | 81.0 | 86.0 | 88.2 | 91.4 |
>     | **DS-GRPO (Ours)** | **62.3** | **70.7** | **76.6** | **81.6** | **86.4** | **89.5** | **92.8** |
>
> * **Outperforming previous methods by balancing diversity and correctness.**
> Our substantial improvement over previous methods stems from a key insight: unlike prior works that focus solely on improving diversity, we are the first to identify and address the fundamental tradeoff between correctness and diversity. We utilize a differential reward function to balance this trade-off, resulting in superior reasoning performance.

---

> ### Author Response · Authors · 2025-11-24
> **Rebuttal Part 2**
>
> ## Question 1: Effect of combining DS-GRPO and entropy regularization (See section F.2 for figures)
>
> >**Question:** Is the improvement additive to entropy regularization if we combine both, or are the effects redundant?
>
>
>
>
>
> **The effect of DS-GRPO is redundant to entropy regularization.**
>
> Thank you for raising this valuable question. The short answer is that the effect of DS-GRPO is redundant to entropy regularization. If we can gain improvement by combining entropy regularization with DS-GRPO, we can tune the parameters of DS-GRPO to achieve better performance.
>
> * **We supplement experiments to confirm our statement.**
>     We first conducted an experiment to verify our conclusion. We added entropy regularization to DS-GRPO on the Countdown dataset using the Qwen2.5-3B-Instruct model. Adding entropy to DS-GRPO does show improvement over vanilla GRPO. However, when we remove the entropy regularization and tune the parameters of DS-GRPO (increasing $\gamma_p$ and decreasing $\gamma_n$), we found that the tuned DS-GRPO still improves upon vanilla GRPO and also outperforms DS-GRPO combined with entropy regularization. This shows that if we can gain improvement by combining entropy regularization with DS-GRPO, we can instead tune the parameters of DS-GRPO to achieve better performance.
>
>     | Method | k=1 | k=2 | k=4 | k=8 | k=16 | k=32 | k=64 |
>     | :--- | :--- | :--- | :--- | :--- | :--- | :--- | :--- |
>     | GRPO | 60.91 | 67.00 | 70.70 | 73.45 | 75.93 | 78.06 | 80.21 |
>     | DS-GRPO | 65.25 | 71.23 | 74.46 | 76.85 | 78.80 | 80.51 | 82.03 |
>     | DS-GRPO + Entropy | 62.35 | 68.00 | 71.97 | 75.06 | 77.66 | 80.03 | 82.52 |
>     | **Diff (DS - Ent)** | **+2.90** | **+3.23** | **+2.49** | **+1.78** | **+1.14** | **+0.48** | **-0.49** |
>
> * **Explanation for the redundancy of DS-GRPO and entropy regularization.**
>     We now focus on explaining why the effect of DS-GRPO is redundant to the effect of entropy regularization.
>     1.  **DS-GRPO can be reformulated as a form of entropy control.**
>         As mentioned in our paper (section on comparing DS-GRPO with entropy control), DS-GRPO can be reformulated as a form of differential entropy control, where we add entropy for the positive trajectories on the loss and subtract entropy for the negative trajectories. Thus, adding entropy regularization can also be reformulated as changing the parameters of DS-GRPO.
>     2.  **The underlying principle by which both DS-GRPO and entropy regularization improve Pass@K is the same.**
>         Another reason why the effect of DS-GRPO is redundant to the effect of entropy regularization is that the underlying principle by which both improve Pass@K is the same: balancing the trade-off between diversity and correctness. Specifically, adding entropy improves diversity at the expense of correctness, while subtracting entropy improves correctness at the expense of diversity. If we can gain improvement by adding entropy regularization over DS-GRPO, it implies that currently the advantage brought by improving diversity outweighs the harm brought by decreasing correctness. Therefore, we can simply tune the parameters of DS-GRPO to obtain a similar effect.
>
>
>
> [1] Song, Yuda, Julia Kempe, and Remi Munos. "Outcome-based exploration for llm reasoning." arXiv preprint arXiv:2509.06941 (2025).
>
> [2] Walder, Christian, and Deep Karkhanis. "Pass@ K Policy Optimization: Solving Harder Reinforcement Learning Problems." arXiv preprint arXiv:2505.15201 (2025).
>
> [3] He, Andre Wang, Daniel Fried, and Sean Welleck. "Rewarding the unlikely: Lifting grpo beyond distribution sharpening." Proceedings of the 2025 Conference on Empirical Methods in Natural Language Processing. 2025.
>
> [4] Dang, Xingyu, et al. "Weight ensembling improves reasoning in language models, 2025." URL https://arxiv. org/abs/2504.10478.
>
> [5] Zhu, Xinyu, et al. "The surprising effectiveness of negative reinforcement in LLM reasoning." arXiv preprint arXiv:2506.01347 (2025).

---

### Official Review · Reviewer_KQde · 2025-11-07

**Soundness:** 3
**Presentation:** 3
**Contribution:** 3
**Rating:** 6
**Confidence:** 3

**Summary:**

This paper analyzes existing methods for combating the loss of large language model (LLM) generation diversity after reinforcement learning (RL)-based post-training. The authors observe that previous methods which maximize generation entropy to improve diversity cause a drop in accuracy, while methods which do the opposite improve the accuracy of the model but harms response diversity. These effects are explained in the paper through theorems that show a "sharpening effect" inherent in traditional RL methods in which correct responses are both more likely to be reinforced, as well as generate greater model updates when they are sampled. The authors then propose their method, DS-GRPO, a modification to the existing GRPO framework which promotes diversity in correct responses while boosting the training signal for negative responses.

**Strengths:**

1. The paper provides a unifying theoretical framework for existing methods which add an entropy-modifying term to the RL objective
2. The empirical results are strong, showing consistent improvement against previous methods
3. Theorems clearly demonstrate the gap that previous methods did not address

**Weaknesses:**

1. In (Section 3.1, line 155 to 161), the paper presents a reward function that is the indicator function for sampled positive examples from $\pi_\text{base}$. It is unclear how applicable this reward function is in practice. Learned reward functions (e.g. from preference data as in RLHF [1]) do not match this framework. When using verifiable rewards as in GRPO [2], the verifier is run on the model outputs in each iteration rather than "sampled from the base policy, $\pi_\text{base}$" (Section 3.1, line 159)

Nitpicks:
2. In (Abstract, line 21), the sentence "...existing heuristics..." should specify comparison to "Vanilla RL and RL with direct entropy maximization" (Section 6, line 439)
3. In (Equation 1, line 150), I did not immediately recognize $\pi^*_\text{van}$ as "the vanilla policy". I recommend using the full word instead.

References:
[1] Ouyang, L. et al. (2022). Training language models to follow instructions with human feedback. https://arxiv.org/abs/2203.02155
[2] Shao, Z., Wang et al. (2024). DeepSeekMath: Pushing the Limits of Mathematical Reasoning in Open Language Models. https://arxiv.org/abs/2402.03300

**Questions:**

1. How are $\gamma_p$ and $\gamma_n$ chosen in the experiments? Can they be optimized separately?
2. How sensitive is DS-GRPO to changes in these hyperparameters?

---

> ### Author Response · Authors · 2025-11-24
> **Rebuttal Part 1**
>
> We thank reviewer KQde for the valuable comments and positive feedback on our work. Below, we would like to address each of your concerns:
> ## Weakness 1: Formulation of reward function
>
> > **Weakness 1:** It is unclear how applicable this reward function is in practice. Learned reward functions do not match this framework. When using verifiable rewards as in GRPO, the verifier is run on the model outputs in each iteration rather than "sampled from the base policy,"
>
> Thank you for raising this insightful question. We believe this point deserves further clarification and discussion.
>
> * **Our theoretical model serves as a simplified abstraction of verifier-based RL algorithms (e.g., GRPO).**
>
>
>
> In practice, verifier-based RL algorithms typically maintain two policies: a sampling policy and a learned policy. The training process is iterative: the model uses a fixed sampling policy to generate trajectories for $t$ iterations, updating the learned policy at each step. After $t$ iterations, the sampling policy is updated to match the current learned policy. Our theoretical model abstracts the learning dynamics within these $t$ iterations where the sampling policy is held fixed. What we demonstrate is that within each such phase, the distribution over correct answers is sharpened.
>
> * **Our theoretical model explains the sharpening observed in practical verifier-based RL training.**
>
> Although our formal analysis focuses on the specific phase where the sampling policy is fixed, the full training process can be viewed as a cumulative composition of these phases. Since the model tends to sharpen the answer distribution within each iteration window (as shown in our theory), the aggregate effect over the entire learning process inevitably leads to a globally sharpened distribution over answers.
>
> * **Our theory also provides intuition for sharpening in RL with other reward functions (e.g., RLHF).**
>
> While our work primarily addresses RL with verifiers, our insights extend to settings with learned reward models, such as RLHF. Specifically, our theory highlights that since the model samples answers from a base distribution, high-probability correct answers are sampled—and thus reinforced—more frequently. A similar mechanism applies to RLHF. During the training of the reward model (or the policy based on it), the system often relies on samples from the base model. Between two equally favorable responses, the one with a higher initial sampling probability is likely to be exposed more often, leading the model to preferentially favor and amplify it. Thus, the "rich get richer" dynamic contributes to sharpening in these settings as well.
>
>
> * **Our empirical results are in practical setting where the verifier is run on outputs in each iteration**
>
> Our empirical implementation adopts the RLVR framework, where the verifier evaluates model outputs at each iteration. Our method is derived directly from our theoretical analysis. These results confirm that our theory provides valuable insights and guidance for practical settings, even when the verifier is applied to outputs in every iteration.
>
>
>
> Thank you again for pointing this out. We have added the above clarification to the revised version of our paper.

---

> > ### Author Response · Authors · 2025-11-24
> > **Rebuttal Part 2**
> >
> > ## Weakness 2: Modification of the presentation in the abstract
> >
> > >**Weakness 2:** the sentence "...existing heuristics..." should specify comparison to "Vanilla RL and RL with direct entropy maximization" (Section 6, line 439) 3. In (Equation 1, line 150), I did not immediately recognize as "the vanilla policy". I recommend using the full word instead.
> >
> >
> > Thank you very much for your advice on presentation and writing. We have addressed this point in the revised version according to your suggestion
> >
> > ## Question 1: Choosing and optimizing $\gamma_n$ and $\gamma_p$
> >
> >
> > >**Question 1:** How are $\gamma_n$ and $\gamma_p$ chosen in the experiments? Can they be optimized separately?
> >
> > **We optimize the hyperparameters $\gamma_n$ and $\gamma_p$ together rather than independently.**
> >
> > Specifically, we employed a two-step procedure:
> >
> > 1. **Step 1: Preliminary range identification (Coarse Search).**
> >    First, we determined a feasible interval for $\gamma_n$ and $\gamma_p$ through a coarse search. By fixing one parameter and varying the other, we observed that excessively large values for $\gamma_n$ or $\gamma_p$ led to training instability and significant performance degradation. Consequently, we established a rough search interval $[a_n, b_n] \times [a_p, b_p]$ within which the training remained stable.
> >
> > 2. **Step 2: Fine-grained selection via Grid Search**
> >    After defining the coarse intervals, we performed a fine-grained grid search to pinpoint the optimal combination. We constructed a two-dimensional uniform grid over the identified intervals and evaluated the candidate combinations:
> >    $\bigg\\{\bigg(\gamma\_n=\frac{i}{N(b\_n-a\_n)},\gamma\_p=\frac{j}{N(b\_p-a\_p)}\bigg)\bigg\\}\_{i,j\in[N]}$
> >    We then selected the parameter set that achieved the best performance on the Pass@K and Pass@1 metrics. It is worth noting that the performance of DS-GRPO is relatively robust to hyperparameter variations within this effective range (please refer to the sensitivity analysis in the following part).
> >
> > ## Question 2: Sensitivity of DS-GRPO to $\gamma_n$ and $\gamma_p$ (see section F.1 for figures)
> >
> >
> > > **Question 2:** How sensitive is DS-GRPO to changes in these hyperparameters?
> >
> > **DS-GRPO is not sensitive to hyperparameters**
> >
> > We constructed a two-dimensional uniform grid over the identified intervals and evaluated the candidate combinations. Our results are shown as follows. We have also added a visualization of the results for different parameters in the revised version of our paper (see section F.1). We can observe that the performance of DS-GRPO on neither Pass@1 nor Pass@K is sensitive to parameters $\gamma_n$ and $\gamma_p$. Based on this observation, it is easy to find the best parameter combination with the method described above. Notably, when $\gamma_p$ is held fixed, adjusting $\gamma_n$ by 0.02 causes Pass@1 and Pass@64 to vary by a narrow margin of approximately 1-2%. These results demonstrate that DS-GRPO remains robust, surpassing the vanilla GRPO baseline across a wide spectrum of hyperparameter settings.
> >
> > Furthermore, given the hyperparameter insensitivity of DS-GRPO, the computational resources and time required for parameter tuning are significantly reduced. This offers a distinct efficiency advantage over baselines and other diversity-oriented algorithms that often demand extensive search processes.
> >
> > | $\gamma_p$ | 0.02 | 0.02 | 0.02 | 0.03 | 0.03 | 0.03 | 0.04 | 0.04 | 0.04 |
> > | :--- | :---: | :---: | :---: | :---: | :---: | :---: | :---: | :---: | :---: |
> > | **$\gamma_n$** | **0.005** | **0.020** | **0.040** | **0.005** | **0.010** | **0.030** | **0.005** | **0.020** | **0.040** |
> > | **Pass@1** | 0.669 | 0.583 | 0.608 | 0.637 | 0.652 | 0.637 | 0.568 | 0.599 | 0.593 |
> > | **Pass@64**| 0.841 | 0.734 | 0.824 | 0.807 | 0.820 | 0.807 | 0.820 | 0.782 | 0.814 |
> >
> > **Table: Pass@1 and Pass@64 under different $\gamma_p$ and $\gamma_n$ settings.**

---

### Author Response · Authors · 2025-12-03
**Official Comment to ACs**

Dear ACs,

We are writing to inform you that we believe we were able to address the concerns of all reviewers.

**Notably, the only reviewer who initially assigned a negative score (4) has acknowledged our response and got a chance to raise their score to a positive 6 before the OpenReview incident**
**(scores change: (8/6/6/4) $\rightarrow$ (8/6/6/6).)**

Specifically, we briefly summarize the concerns of the reviewers and how we address them as follows:


## Reviewer KQde
>**Question: Applicability of reward function in practice**

  *Answer:* **We clarify that our analysis applies to verifier-based RL and serves as an abstraction of methods like GRPO.** In these methods, the sampling policy is held fixed for a number of steps and then iteratively updated. Our analysis (which assumes a fixed sampling policy) remains valid for each individual step of this process, where the sharpening effect observed at each step accumulates over iterations.


>**Question:** Is our method sensitive to hyper-parameters, and how do we choose them?

 *Answer:* **Our method is robust to hyper-parameter variations. We selected the hyper-parameters via a standard two-dimensional grid search.** We have also validated this claim with additional experiments.

## Reviewer fC3U
>**Question:** Reviewer was positive about the conistent gains but expressed that they were modest.

  *Answer:* **We contextualize our gains in the context of other recent works that also focus on improving pass@K after RL on LLMs.** Our method's improvement (6.7%) is in fact the highest among these works that look at comparable settings. Hence, we believe our method presents a significant advancement, especially in light of the consistent gains.

## Reviewer 17fN
>**Question:** Can DS-GRPO outperform CISPO?

*Answer:* **We have conducted additional experiments to show that our method outperforms CISPO.** Specifically, we achieve up to a 10.0% improvement in Pass@64 on the AIME25 dataset.

>**Question:** Does DS-GRPO exhibit higher diversity compared to the base model?

   *Answer:* **We conduct additional experiments to demonstrate that DS-GRPO achieves higher diversity than the base model.** Following the evaluation method suggested by Reviewer 17fN, we compared the diversity of the base model, GRPO, and DS-GRPO. Our results show that GRPO has lower diversity than the base model, whereas DS-GRPO exhibits higher diversity than the base policy.

>**Question:** Does differential entropy control outperform other entropy control methods?
>
   *Answer:* **We show with additional experiments that differential entropy control outperforms other entropy control methods.**

The reviewer expressed that they were happy with our rebuttal and raised their score.
## Reviewer jhoN
>**Question:**  The authors should calculate the average uplift between DS-GRPO and GRPO over model families for each dataset.

*Answer:* **We present the results averaged over model families and datasets, demonstrating that our improvement is statistically significant.**

>**Question:**  The effect of entropy penalty and bonus is not obvious on the Countdown task and could just be noise.

 *Answer:* **We performed experiments across multiple random seeds to validate the reliability of our results.** We confirm that the performance trends comparing entropy regularization (penalty/bonus) with GRPO are robust to stochastic variations and statistically significant across different datasets.

>**Question:**  Has the sharpening effect been observed in RL prior to LMs?

*Answer:* **We have listed several references confirming that the sharpening phenomenon was observed in RL literature prior to LMs.**

---

### Meta-Review · Area_Chair_ovMa · 2025-12-23

**Summary:**

Overall reviewers agree this is good contribution. The main concerns affecting the decision were (i) how well the theory matches practical  training setups, (ii) whether reported gains are meaningfully beyond variance (iii) missing comparisons and ablations, and (iv) limited scaling evidence. The rebuttal and added experiments have mostly addressed the main concerns against publication. The only reviewer who initially assigned a negative score (4) has acknowledged the response and raised score to 6. Updated scores (8/6/6/6) are.

However, in my own reading I found an important issue affecting the correctness and/or substantiation of the main theoretical claim (Theorem 6.1). While this does not necessarily invalidate the algorithmic contribution, it undermines the paper’s strongest claims about the scope and precision of the theoretical justification (e.g., the abstract’s “Our theory precisely characterizes when existing heuristics help and why they fail, while showing that differential smoothing is universally superior.”). I therefore recommend the authors revisit and correct the theory prior to publication, and/or substantially tone down the related claims.

**Main issue:**

In Defn 2, diversity \sigma as defined is a **non-positive** quantity since $\sum_i \pi_i^2 \leq (\sum_i \pi_i )^2$ for non-negative $\pi_i$
The main theorem Thm 6.1 claims that their method has higher diversity than the vanilla method, i.e. $\sigma(\pi_{DS})>=\sigma(\pi_{ent}).$ Two issues with this:

(a) **Interpretation / direction mismatch.** given that \sigma is non-positive, this means that \sigma(\pi_DS) is closer to zero. But under the definition above, being closer to 0 corresponds to more concentration of mass on a smaller number of correct trajectories (indeed, the maximum 0 occurs when all correct mass is on a single trajectory). This contradicts the paper’s interpretation that DS yields a more diverse correct-solution distribution. If the intent was to claim greater diversity under this metric, the inequality direction appears reversed (or the definition should use the opposite sign / an equivalent nonnegative diversity measure).

(b) **Missing Proof.** Unfortunately, I could not check if this issue is attributed to perhaps a typo in the definition of \sigma since I cannot find any proof of this claim in the appendix (App. B.3). Thm B.1 in the appendix restates the claim $$\sigma(\pi_{DS})>=\sigma(\pi_{ent})$$ Its proof claims it follows from lemmas 4-7 but these are all statements about C (first statement of theorem) not \sigma (second statement of theorem).

**smaller issues:**

Proof of the 6.1 in appendix is given as a list of lemmas that all look similar and seems would easily admit some unification of the one idea that seems to be repeated. This not only makes difficult to read and the text extremely repetitive but also results in typos by repeated arguments eg. last two lines of proof of Lemma 6 refer to KL divergence while lemma is about chi-square divergence. End of lemma 7, again mention to KL divergence.

The selection biased statement Prop B.1 is based on a “definition”/assumption that: a correct trajectory receives a positive reward only if it is sampled within N rollouts. This is then combined with another assumption (ignoring possibly coupled optimization dynamics between trajectories) that the likelihood of a correct trajectory increases. In particular, it is not proved that these two assumptions hold for the definition of \pi_{base}^* in Eq. (1) which is what Prop B.2 actually uses.

**Reviewer Concerns:**

Addressed:
* added key baselines/ablations
* stronger empirical support (more seeds/averages)
* hyperparameter/sensitivity details

outstanding:
* limited scaling evidence beyond ~7B (mild)

**Reviewer Scores:**

I believe final scores would have changed from 8/6/6/4 to 8/6/6/6. The change of 4 to 6 was confirmed by the reviewer after authors' response.

---

### Decision · Program_Chairs · 2026-01-26

Reject